# Evaluation of Human-AI Teams for Learned and Rule-Based Agents in Hanabi

**Ho Chit Siu** [*]     **Jaime D. Peña** [*]     **Yutai Zhou** [*]     **Edenna Chen** [†]     **Victor J. Lopez** [‡]

**Kyle Palko** [‡]               **Kimberlee C. Chang** [*]               **Ross E. Allen** [*]

## Abstract

Deep reinforcement learning has generated superhuman AI in competitive games such as Go and StarCraft. Can similar learning techniques create a superior AI teammate for human-machine collaborative games? Will humans prefer AI teammates that improve objective team performance or those that improve subjective metrics of trust? In this study, we perform a single-blind evaluation of teams of humans and AI agents in the cooperative card game *Hanabi*, with both rule-based and learning-based agents. In addition to the game score, used as an objective metric of the human-AI team performance, we also quantify subjective measures of the human's perceived performance, teamwork, interpretability, trust, and overall preference of AI teammate. We find that humans have a clear preference toward a rule-based AI teammate (SmartBot) over a state-of-the-art learning-based AI teammate (Other-Play) across nearly all subjective metrics, and generally view the learning-based agent negatively, despite no statistical difference in the game score. This result has implications for future AI design and reinforcement learning benchmarking, highlighting the need to incorporate subjective metrics of human-AI teaming rather than a singular focus on objective task performance. [4]

## 1  Introduction

Advances in artificial intelligence (AI) have resulted in agents that perform at superhuman levels within domains previously thought to be solely the purview of human intelligence. Such performance has been demonstrated through application of reinforcement learning (RL) in environments such as board games [6, 39], arcade games [31], real-time strategy games [44], multiplayer online battle arenas [4], and simulated aerial dogfights [11].

In nearly all cases, these demonstrations of AI superiority are in purely adversarial, one- or two-player games. However, in order to achieve real-world applicability and adoption, AI must be able to demonstrate *teaming intelligence*, particularly with human teammates [23]. Due to the focus on adversarial games, teaming intelligence has been understudied in RL research. AI teammates must also exhibit behavior that engenders an appropriate level of certain human reactions, such as trust, mental workload, and risk perception [26, 37]. Failure to do so risks the same kind of misuse, disuse, and abuse that Parasuraman and Riley illustrated with traditional automation systems [37]. These issues are distinct from much of current multi-agent AI work, as AI that is able to team effectively with other AI agents has failed to work effectively with humans [9].

---

[*]MIT Lincoln Laboratory, `{hochit.siu,jdpena,yutai.zhou,chestnut,ross.allen}@ll.mit.edu`
[†]MIT Department of Electrical Engineering and Computer Science, `edenna@mit.edu`
[‡]U.S. Air Force Artificial Intelligence Accelerator, `{victor.lopez.10,kyle.palko.1}@us.af.mil`

[4]DISTRIBUTION STATEMENT A. Approved for public release. Distribution is unlimited.

35th Conference on Neural Information Processing Systems (NeurIPS 2021)

The objective of this paper is to evaluate human-AI teaming in the cooperative, imperfect-information game of Hanabi. We consider not only the *objective* performance of a human-AI team, but also the *subjective* human reactions and preferences when working with different kinds of AI teammates. Based on the success of applying deep reinforcement learning to create superhuman AI in adversarial games, we hypothesize that similar RL techniques can render collaborative AI that outperform and are preferred over rule-based agents in human-AI Hanabi teams. Our results show that this hypothesis is *not* supported given the current state of the art of collaborative RL agents. Human participants show a clear preference toward rule-based AI even though the learning-based AI perform no worse and are specifically optimized for teaming with previously unknown partners (e.g. humans) [21]. To the best of our knowledge, this is the first comparative study of objective performance of rule-based and learning-based Hanabi AI in human-teaming experiments, as well as the first quantified study of subjective human preferences toward such AI.

## 2 Background

### 2.1 Hanabi

*Hanabi* is a cooperative card game in which two to five players attempt to stack twenty-five cards into five different fireworks (piles), one for each suit (color) and by ascending rank (number). We consider only the standard, two-player version, where the deck is composed of 50 cards, five suits, each suit having three 1s, two 2s, two 3s, two 4s, and one 5, and each player is dealt a hand of five cards. Hanabi's difficulty lies in the fact that players can only see their teammate's hand and never their own and communication about cards is strictly limited.

Games start with eight *hint tokens* and three *bomb tokens*. Each turn, a player may *discard* a card from their hand, *play* a card from their hand, or give a *hint* to a teammate. Whenever a card is played or discarded, it is revealed and a new card is drawn from the deck. A correctly-played card is placed in the appropriate firework; an incorrectly-played card is discarded and the team loses one bomb. Hinting costs one hint token and allows a player to reveal either all cards of a certain suit or rank in their teammate's hand. Hint tokens are earned back when a card is discarded, or when a 5 card is successfully played. The final score is the sum of the top card in each firework, for a maximum of 25 points. The game ends when all fireworks have been completed, the deck is empty (and each player has one additional turn), or the team loses 3 bombs.

Hanabi is a *purely cooperative* game with imperfect information, and limited, codified communication. These properties make it an interesting challenge for teaming since players must consider the reasons for their teammates' actions and any implied information, while avoiding misinterpretations. Bard et al. [2] presents a more complete treatment of Hanabi and its properties as an AI problem.

### 2.2 Training and Evaluating AI Teams

Hanabi's uniquely collaborative nature has made it the subject of several AI challenges in recent years [2, 46]. In these challenges AI agents are paired with other teammate agents, and their performance is measured based on the paradigms of *self-play* and *cross-play* [21].

Self-play (SP) is when an AI plays a game with a copy of itself as the opponent (adversarial games) or teammate (cooperative games like Hanabi). Self-play can be used as a form of RL training [1, 20], and/or as a form of evaluation [2].

Cross-play (XP) is an evaluation-only paradigm where an agent is teamed with other agents (AI or humans) that were not encountered during training, measuring how well an agent can cooperate with previously-unseen teammates. SP-trained agents can achieve high SP-scores by developing "secretive conventions" that are not understood by agents not present during training, thus completely failing in the XP setting [21]. Cross-play with humans (*human-play*) is of particular importance as it measures human-machine teaming (HMT) and is the foundation for the experiments in our paper.

Cross-play can be evaluated under the zero-shot coordination (ZSC) [21] or ad-hoc teaming [40] settings. ZSC assumes that teams are formed between agents that have no prior knowledge of each other; therefore it is impossible to train an agent to bias towards teammate idiosyncrasies, though the agents may agree to a common training strategy beforehand. On the other hand, ad-hoc teaming attempts to achieve coordination by having agents update their policy while interacting with other

Table 1: Survey of Rule-based and Learning-Based Hanabi AI Performance in 2-Player Games

| Hanabi AI | Original Author | References | Self-Play | Cross-Play | Human-Play |
|---|---|---|---|---|---|
| Van den Bergh | Van den Bergh 2016 | [42, 45, 15] | 13.8 | 10.8 | – |
| Self-Recognition | Osawa 2015 | [36, 15] | 15.9 | – | – |
| Piers | Walton-Rivers 2017 | [45, 38] | 17.3 | 11.2 | – |
| Intentional AI | Eger 2017 | [13, 8] | 12.6 | **13.8** | **15.0** |
| Expectimax | Bouzy 2017 | [5, 15] | 19.0 | – | – |
| MirrorSituational | Canaan 2018 | [7, 15] | 20.1 | 12.4 | – |
| RIS-MCTS | Goodman 2019 | [15, 8] | 20.6 | 13.3 | – |
| WTFWThat | J. Wu 2018 | [48, 2, 15] | 19.5 | – | – |
| FireFlower | D. Wu 2018 | [47, 2, 14] | 22.7 | – | – |
| Implicature AI | Liang 2019 | [28, 8] | 18.9 | – | $\sim$ **15** |
| **SmartBot (SB)** | O'Dwyer 2019 | [33, 2, 15] | **23.0** | – | – |
| IL-Valuebot | Sarmasi 2021 | [38, 33] | 18.0 | – | – |
| ACHA | Bard 2020 | [2, 20] | 22.7 | 1.0 | – |
| BAD | Foerster 2019 | [14, 2, 20] | 23.9 | – | – |
| SAD | Hu 2019 | [20, 21] | 24.0 | 3.0 | – |
| SAD+AUX | Hu 2019 | [20, 21] | 24.0 | 21.1 | 9.2 |
| SPARTA | Lerer 2020 | [27, 8] | **24.6** | – | – |
| **Other-Play (OP)** | Hu 2020 | [21] | 24.1 | **22.5** | **15.8** |

We provide Hanabi AI agents' scores as they are reported in existing literature. This gives a notional perspective of the current state of the art; however, we note that evaluation conditions have not been standardized throughout previous literature (i.e. number of game seeds for self-play, pool of teammates for cross-play, etc.). Therefore some caution is needed when making direct comparisons between reported scores. Please see respective literature for details on evaluation conditions.

agents, and the the pool of partner agents can be known in advance (though it can be of arbitrarily large size, making such knowledge practically impossible to exploit).

## 2.3 AI for Hanabi

Table 1 summarizes the most salient published Hanabi AI agents in 2-player games with game score in self-play, cross-play, and human-play. The table is separated into *rule-based* (top) and *learning-based* (bottom) agent types. Rule-based agents have a policy composed of a predefined set of rules to follow given any particular game situation, and the rules are often derived from human domain knowledge. Learning-based agents, on the other hand, use statistical learning methods to adjust the parameters of their policy. The mechanism governing what action the policy will choose is learned via the agent's experience, without the need for human domain knowledge.

Most early work focused on rule-based AI [36, 42, 45]. Goodman won the 2018 CoG Hanabi Competition [46] with a rule-based Monte Carlo Tree Search agent. O'Dwyer's SmartBot (SB) is the highest performing rule-based agent created to date, in terms of self-play [33, 2, 15]. Due to its state-of-the-art SP score, as well as a readily available implementation [33], we select SB as our rule-based agent in the human-AI experiments presented in Sections 3 and 4. We found no previous work that evaluated SB in the cross-play or human-play settings.

Recently developed learning-based agents have demonstrated breakthrough Hanabi performance. Sarmasi et al. provide a collection of agents trained with imitation learning that nearly match the performance of the rule-based agents from which they were trained [38]. A sequence of publications [14, 20, 27, 21] offered reinforcement learning agents that each advanced the state of the art self-play and/or cross-play performance at the time of publication. This culminated with the SPARTA [27] and Other-Play (OP) [21] agents; which, respectively, represent the highest performance to date in self-play and cross-play of any agent type.

The OP agent is selected as the learning-based agent for study in our human-AI experiments described in Sections 3 and 4. We choose OP, not only for its state-of-the-art cross-play performance, but also because it uses a learning objective function specifically designed to optimize for zero-shot coordination settings. Due to this optimization, we expect that humans would subjectively prefer and

objectively perform better with OP agent teammates over other AI teammates that were not designed for the zero-shot coordination setting, such is SmartBot.

A handful of works have conducted human-play experiments; however these works are uncommon due to the significant time and effort required to conduct such experiments. Both Eger et al. [13] and Liang et al. [28] propose rule-based AI derived from Gricean maxims [17] and achieved an average human-play score of approximately 15.0 over a set of experiments that included hundreds of human participants. The learning-based Other-Play agent is arguably the highest performer in terms of human-play with an average score of 15.8, however the experiments were run with a much smaller pool of human participants [21]. No prior works were found that provide a comparative study of rule-based and learning-based Hanabi AI in human experiments.

## 2.4 Human-AI Teaming and Metrics

Human-machine interaction evaluations typically consider two broad categories of metrics: objective performance metrics (raw score, error rates, accuracy, time required, etc.), and subjective team- or human-focused metrics (situation awareness, trust, workload, etc.). Measurement of the former is heavily task-dependent, and are often the primary evaluation metrics of AI systems. The latter, however, can also involve metrics for probing systems and teams in a quantitative way, with important implications for how technology is used and adopted [37].

One key metric of teaming is *trust*, which is defined by Lee and See as "the attitude that an agent will help achieve an individual's goals in a situation characterized by uncertainty and vulnerability" [26]. Potential difficulties with trust include trust *calibration* (whether one's trust of an agent is commensurate with its capabilities) and trust *resolution* (whether the range of situations where a human trusts a system is commensurate with its range of capabilities).

Closely related to trust are the notions of *legibility* (being expressive of one's intent) and *predictability* (matching one's expectations) [12]. In measuring teaming, one might directly ask humans about their teammates' intent and their own expectations, particularly in relation to shared goals. Dragan et al. [12] argue that these two ideas trade off in the context of robot motion, but that legibility is the more important factor when working in close collaboration with a human. Similar arguments may also be made in the context of Hanabi, where maximum legibility is key to guiding a team's actions in the tightly-coupled, imperfect-information scenario.

Hoffman [19] proposes a set of subjective and objective methods (overlapping with the aforementioned ideas) to measure *fluency* in human-robot interaction, defined as the "coordinated meshing of joint activities between members of a well-synchronized team." Fluency encompasses subjective elements such as teamwork, trust, and positive perception; and objective measures of timing and concurrency.

Much of the contemporary work on learning-based agents has not evaluated such teaming metrics, due to their focus on single-player [3] or adversarial games [39, 43]. In this work, we consider the use of both objective performance and subjective teaming-based measurement of human-AI pairs in the context of Hanabi. We aim to provide a more thorough analysis of the human-AI teaming data than previously-discussed Hanabi HMT studies [13, 28, 21]. Rather than proposing a new AI with better raw performance, we are interested in the subjective aspects of human-AI teaming with state-of-the-art rule- and learning-based AI, and how they might drive future AI development.

# 3 Methods

## 3.1 Human-AI Teaming Experiment

Experiments consisted of two-player Hanabi games played by teams of one human participant and one AI agent. Experiments aimed to measure the objective team performance and subjective human reactions to different types of AI. The AI agents used in experiments were the Other-Play RL agent [21] (specifically the *OP+SAD+AUX* agent, hereafter referred to as the *OP* bot) and the "SmartBot" agent [33] (hereafter referred to as the *SB* bot), both of which have MIT Licenses. These agents were chosen because they were the top-performing learning-based and rule-based Hanabi AI, respectively at the time of the experiment.

Participants were first introduced to the experiment and the rules of Hanabi as defined in Section 2.1. The experiment followed with a brief familiarization game so participants could acquaint themselves with the game interface. Then, each participant played two sets (blocks) of three games, with each set using a different AI teammate. The participant was not informed which AI teammate played each block of games and the order of the AI teammates was counterbalanced over the course of the study. Participants answered Likert scale surveys after each game (Table 2, left), a NASA Task Load Index (TLX) survey after each block [18], and a Likert scale survey after both blocks that directly compared their experience with both agents (Table 2, right). Likert scale survey questions were largely derived from a compilation of similar questions in Hoffman et al. [19]. Unless otherwise noted, Likert scales were arranged with 1 corresponding to "strongly disagree" and 7 corresponding to "strongly agree."

A total of 29 adult participants completed an experiment and each provided written, informed consent. The protocol was approved by the MIT Committee on the Use of Humans as Experimental Subjects (protocol E-2520) and the United States Department of Defense Human Research Protection Office (protocol MITL20200003). Participants received a $10 USD gift card at the end of their experiment, and the highest-scoring participant received an additional $50 gift card. Each experiment took approximately 1.5 to 2.5 hours. The total amount spent on participant payment was $350, with one participant being paid twice due to technical difficulties ending their initial session early. Experiments were conducted virtually, with all interactions occurring through video-conference, online surveys, and the Hanabi game interface, adapted from [27]. No personally-identifiable information was collected, and no significant risk to participants was expected.

Liang et al. [28] conducted human experiments with subjective survey questions; however the only significant result reported was that humans were more likely to mistake Implicature AI as another human. Although Hu et al. [21] conducted some human experiments with the OP bot and a self-play RL bot, their experiments only focused on game score, involved only one game with each agent, and did not consider participant expertise in Hanabi.

Table 2: Post-Game (Left) and Post-Experiment (Right) Evaluation Statements

|     | 7-point Likert Scale statement |
| --- | --- |
| G1 | I am playing well. |
| G2 | The agent is playing poorly. |
| G3 | The team is playing well. |
| G4 | This game went well. |
| G5 | The agent and I have good teamwork. *[fluency]* |
| G6 | The agent is contributing to the success of the team. |
| G7 | I understand the agent's intentions. *[legibility]* |
| G8 | The agent does not understand my intentions. *[legibility]* |
| G9 | I feel comfortable playing with this agent. |
| G10 | I do not trust the agent. *[trust]* |
| G11 | The agent is not a reliable teammate. *[predictability]* |
| G12 | I am not confident in my gameplay. |

|     | 7-point Likert Scale statement |
| --- | --- |
| E1 | Which agent did you prefer playing with? |
| E2 | Which agent did you trust more? *[trust]* |
| E3 | Which agent did you understand more? *[legibility]* |
| E4 | Which agent understood you better? *[legibility]* |
| E5 | Which agent was the better Hanabi player? |
| E6 | Which agent was more reliable? *[predictability]* |
| E7 | Which agent had a better understanding of the game on average? |
| E8 | Which agent caused you to have a greater mental workload? *[mental workload]* |

Left: Statement order was randomized when presented to participants.
Right: Rating 1 is "strongly prefer first agent" and rating 7 is "strongly prefer second agent."
Items associated with particular human factors constructs have the construct name in brackets (not shown to participants).

## 3.2 Statistical Analysis

We evaluated both objective and subjective metrics of human-AI teaming, with the overall hypothesis that the Other-Play RL agent (OP) is preferred over- and would outperform the rule-based SmartBot (SB) agent. As an objective measure, we consider the team score during each game. For subjective measures, we consider outcomes related to perceived performance, teamwork, legibility, comfort, trust, and workload as measured by the Likert surveys. Linear mixed-effects regression models were used for both objective and subjective variables, following the recommendation of Norman [32].

Due to space constraints, we do not present the results of the TLX here, but include one question on mental workload from the post-experiment survey (Table 2, E8).

Both objective and subjective omnibus tests for post-game results used second-order mixed-effects models with fixed factors of (1) AI agent, (2) self-rated Hanabi experience, (3) block (the first or second set of three games), and (4) game number (first, second, or third game within a block), and a random factor of participant number. AI agent and participant number were considered categorical variables. Post-experiment surveys were evaluated separately with one-sample $t$-tests, and a Holm–Bonferroni multiple comparison correction, since these tests were looking for differences from a neutral value on the Likert scale, and there were multiple hypotheses being tested. Additionally, correlations between self-rated experience, and post-game responses were checked against game score.

# 4 Results

We present the results of the teaming experiment. Our results indicate that despite *no significant difference in objective performance* between teaming with the two agents (Section 4.1), human sentiment results (Sections 4.2 and 4.3) show a *clear preference toward a rule-based agent over a learning-based agent*.

For context, we have colored most of the plots to show the self-rated Hanabi experience level of the participant with which the data are associated, and we note that our participant pool is skewed towards higher-self-rated-experience players. Self-rated Hanabi experience was based on ratings of the demographic survey statement "I am experienced in Hanabi" (see supplemental materials for demographic survey results).

## 4.1 Game Score

The mixed-effects regression *did not* support score differences due to agent type ($t(158) = 0.374), p = 0.709$) or self-rated Hanabi experience ($t(158) = 1.228, p = 0.221$) (Figure 1). This means that the data do not support OP under-performing relative to SB.

The dependent variable of game score only has a significant effect of block ($b = 5.282, t(158) = 2.636, p = 0.009$), and a near-significant effect of game number ($b = 2.957, t(158) = 1.887, p = 0.061$). The positive regression slope in both cases indicates that this is likely a learning effect over time; i.e. the human participant adapts his/her play over time.

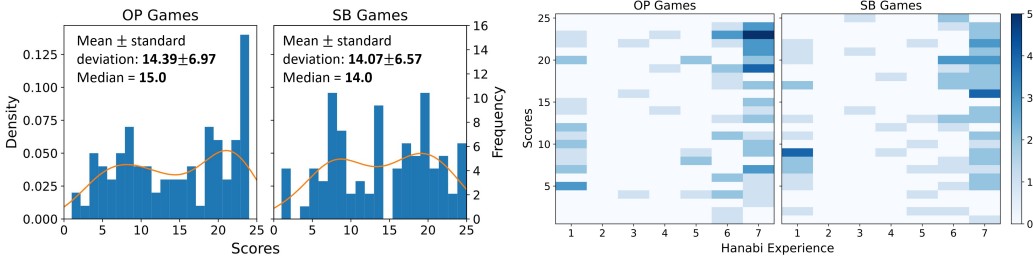

Figure 1: Game scores by agent type (left) and self-rated player experience (right). No significant differences were found when teaming with either agent, and correlation with self-rated experience was significant only for SmartBot games.

Since score is the primary performance metric of interest in Hanabi, and acts as the reward function for RL agents in this domain, we examined a few additional correlations with score. Note that since these are bivariate correlations, they are not as complete as the aforementioned statistical regression.

Self-rated Hanabi experience and score had a small, but significant positive correlation when pooling both agents' games ($p = 0.0053, r = 0.214$). Correlation remains significant for the subset of SB games ($p = 0.0023, r = 0.247$), but is not significant for OP games ($p = 0.0867, r = 0.1881$), indicating that for this bivariate analysis, experience only correlates with score for SB, not with OP.

A small, but significant positive correlation was found between subjective team performance (G3, G4) and score ($p = 0.0003$, $r = 0.275$ and $p = 0.0002$, $r = 0.280$). However, subjective ratings of self- and agent-performance (G1, G2) were not significantly correlated to score.

When participants were ordered by the absolute differences in their total scores with each agent, the ten subjects with the greatest differences (max/min point differences of 57 and 17), six of the ten had better performance with OP, but when considering only the top five subjects in terms of score difference (max/min differences of 57 and 28), only the participant with the greatest difference performed better with SB, and all four others performed better with OP. A full plot of scores by subject and agent type is in the supplemental materials.

## 4.2 Post-Game Sentiments

Significant effects in post-game subjective measures are summarized in Table 3. Although the statistical model considers factors of block and game, which were sometimes significant, we do not report these main effects or their interactions in the table, in order to focus on the independent variables of greater interest.

In all cases where the interaction of agent type and Hanabi experience was found to be significant, more experienced Hanabi players rated the Other-Play agent much more negatively than the SmartBot agent, while novices rated the two similarly (Figure 2). However, there was no significant difference between novice and expert ratings of the SmartBot agent. Cases where neither agent nor experience were significant factors are shown in Figure 3.

Table 3: Post-Game Sentiment Statistics (Statistically-Significant Factors Only)

| Dependent Variable | Factor | $t$ | $p$ |
|---|---|---|---|
| I am playing well (G1) | experience | 2.698 | 0.008 |
| The agent is playing poorly (G2) | experience | 4.044 | $< 0.0001$ |
| The team is playing well (G3) | agent:experience | $-2.082$ | 0.039 |
| The agent and I have good teamwork (G5) | agent | 2.578 | 0.011 |
| | agent:experience | $-3.021$ | 0.003 |
| I understand the agent's intentions (G7) | agent:experience | $-2.273$ | 0.024 |
| The agent does not understand my intentions (G8) | experience | 2.098 | 0.037 |
| | agent:experience | $-3.166$ | 0.002 |
| I feel comfortable playing with this agent (G9) | agent | 3.302 | 0.001 |
| | agent:experience | $-3.561$ | $< 0.0001$ |
| The agent is not a reliable teammate (G11) | experience | 3.159 | 0.002 |

Degrees of freedom for $t$-tests are 164 in all cases. *Agent* is agent type (SmartBot or Other-Play), and *experience* is self-rated Hanabi experience. *Agent:experience* is the interaction effect of agent and experience

To specifically examine the effect of self-rated player experience, we performed post-hoc pairwise comparisons in cases where experience was significant. Participants were pooled into "novice" ($n = 10$, self-rated experience of $\leq 5$) and "expert" ($n = 19$, self-rated experience of $> 5$) groups, and comparisons were made on ratings of each agent in the cases where an interaction effect was significant (G3, G5, G7, G8, G9). The groups did not rate SB significantly differently, but experts always rated OP worse than novices did. The difference in G3 "the team is playing well" ($t(85) = 3.551$, $p < 0.001$, effect size $d = 0.752$) was not as stark as the others ($t(85) = 5.068$ to $5.855$, $p < 0.0001$, $|d| > 1.0$), but all were still clearly significant, and all but one case had large effect sizes.

## 4.3 Post-Experiment Sentiments

Participants' direct comparisons of the agents (Table 2) are shown in Figure 4. The statements were presented to participants as comparing the "first" and "second" agents (without reference to agent type, which was not disclosed to participants). For ease of interpretation, we matched the responses to the agent type, and flipped the scale as appropriate. All positively-framed questions (E1 to E8) showed a strong preference to SB over OP (corrected $p < 0.05$), while mental workload (E8) was split.

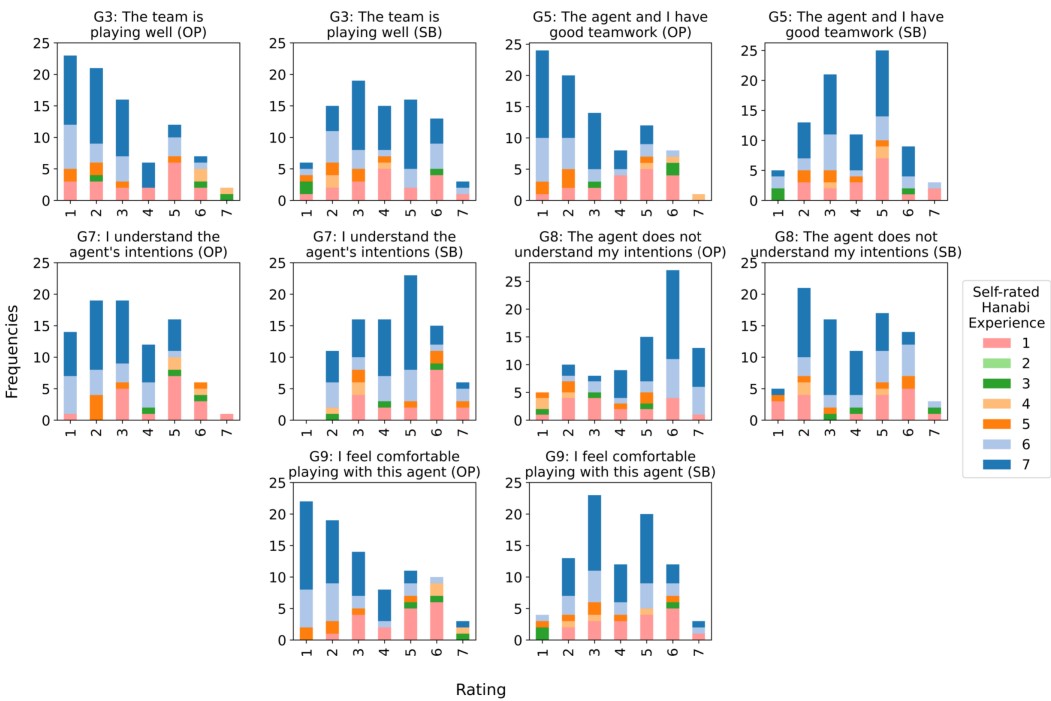

Figure 2: Participant rating for post-game questions by self-rated Hanabi experience (SB vs OP), where statistically significant differences related to factors of agent and/or experience were found (Table 3). The scale ranges from 1-7, corresponding to "strongly disagree" to "strongly agree".

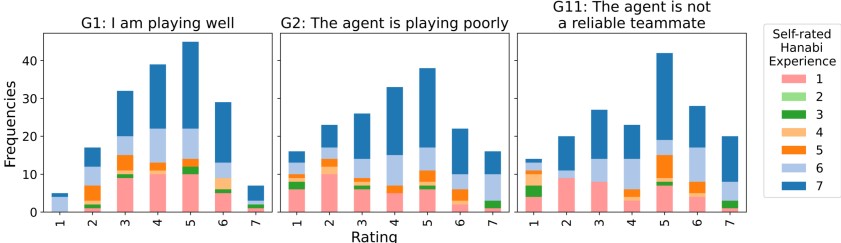

Figure 3: Participant rating for post-game questions by self-rated Hanabi experience where no statistically significant differences related to factors of agent and/or experience were found. The scale ranges from 1-7, corresponding to "strongly disagree" to "strongly agree".

We note there were three participants who obtained a score of 24 with OP, with one of these participants obtaining 24 twice (no player achieved a 25 with OP). All three replied at the extreme end of our Likert scale (i.e., 1 or 7) for question E1 with a preference for SB. Interestingly, their cumulative scores for OP and SB, respectively, were: Participant 6 (played with OP first, self-rated experience of 7): 57 and 28; Participant 19 (SB first, experience 7): 68 and 48; Participant 20 (OP first, experience 6): 70 and 35. The participant with the highest cumulative score (Participant 2, OP first, experience 7) had cumulative scores of 68 (OP) and 54 (SB) and preferred SB with a Likert rating of 6. All participant scores are provided in the Appendix (Figure 7).

Participant commentary indicated that low mental workload when working with OP was often caused by frustration with the agent and giving up on teaming with it. For example, after the OP bot failed to act on several hints from the human (*"I gave him information and he just throws it away"*) a participant commented that *"At this point, I don't know what the point is,"* regarding working with the agent.

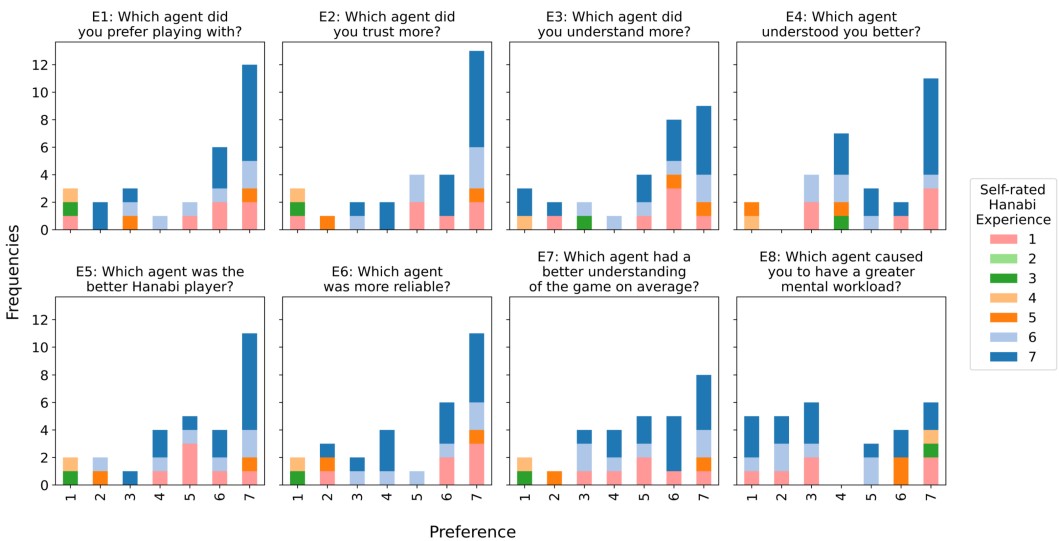

Figure 4: Post-experiment questions. All showed a statistically significant preference for the rule-based SmartBot ($p < 0.05$), except the question on workload (E8). The scale ranges from 1 ("strongly prefer OP") to 7 ("strongly prefer SB").

## 5 Discussion

### 5.1 Other-Play Agent Performance

Other-Play (OP) training is designed to avoid the creation of "secretive" conventions that can result from self-play training. However, even though OP agents can pair well with previously-unseen teammates, there are still assumptions placed on the types of teammates OP agents will encounter. Notably, OP assumes that teammates are also optimized for zero-shot coordination [21]. In contrast, human Hanabi players typically do not learn with this assumption. Pre-game convention-setting and post-game reviews are common practices for human Hanabi players, making human learning more akin to *few-shot* coordination. Furthermore, Hu et al. note that, due to practical implementation details, OP does not always avoid secretive conventions. For example, some OP agents use color hints to indicate discarding of the 5th card [21]. Such conventions are not legible to a human teammate without extensive observation.

### 5.2 Implications for Human-AI Teaming

The results shown here have important implications for the development of learning-based, human-teaming agents. Regardless of the objective performance of AI systems, "teaming intelligence" is ultimately required for real-world deployment. The difference between rule-based and learning-based systems is particularly poignant here, where humans strongly favored a rule-based agent over an RL agent in many subjective metrics (and never significantly favored the RL agent) despite achieving objective scores that were not significantly different across the two agent types. Caution should be exercised when generalizing our results beyond Hanabi, though there is some indication of cross-domain consistency. The negative post-experiment sentiments expressed towards OP by some participants are similar to those expressed by participants during the DotA2 OpenaAI Five games where RL players exhibited playstyles that were not immediately understood by human players, even though they were successful [34, 35].

An additional consideration is the user base of future learning-based systems. We note that between post-game and post-experiment surveys, nearly all statements/questions associated with human factors constructs (except workload) had differences related to the agent and/or the participant's experience level. Experienced players rated the OP agent particularly negatively (Figure 2, Table 3), despite similarity in scores. As domain experts are likely to be the first users of AI technology in operational settings, their perception of AI is a key factor to its adoption. Part of the experts' negativity toward OP may come from the way human experts make decisions in uncertain situations.

Klein's recognition-primed decision-making model [25] indicates that experts typically rely on a base of knowledge to recognize "prototypical" situations and alleviate much of the mental burden of decision-making. Working with an unpredictable and illegible teammate breaks experts' ability to rely on much of their knowledge base. This notion is also supported by the score/experience correlations (Figure 1) which were only significant for SB games.

This work indicates a need for RL methods that produce legible and predictable policies. Methods in this direction generally involve either building inherently transparent models, or procedures for post-hoc analysis. The former includes human-language-driven policy specification [41], or explanation learning through construction of causal diagrams [30]. The latter includes saliency maps for visual domains, and action preference explanation through reward decomposition [24].

### 5.3 Limitations and Future Work

The tight coupling of coordination and performance, and the codification of communication, make Hanabi a unique test bed. Games without these properties may elicit different responses from human-AI teams, so generalization of this experiment beyond Hanabi must be treated carefully.

Due to the manual and time-intensive experimental procedure —as well as the lack of readily available implementations for many pre-existing Hanabi agents such as Intentional [13] and Implicature AI [28]—we were limited in our pool of participants and agents. Future work could increase both by using an online, operator-free game-play and survey platform.

While we made an effort to recruit participant with a diverse level of experience, the participant pool was still skewed towards those with higher self-rating (though it is likely less biased than the pool from [28], who recruited exclusively from online gaming communities). Since Hanabi performance is heavily dependent on the *team*, it is difficult to obtain an objective single-player skill rating. Still, the relatively tight clustering of many (self-rated) experts' responses to OP in particular is notable.

Game outcomes and the feasibility of perfect games (i.e. score 25) are dependent on the starting deck. We did not control for deck configuration to avoid the issue encountered in Hu et al. [21], who explored a relatively narrow portion of the game space due to using only two deck seeds.

Interesting extensions to this Hanabi experiment include varying participants' knowledge of their teammates, increasing the number of games played, and adding additional metrics of teaming, such as positive listening and positive signalling [22, 29]. Modifications to the agents may include adding logic filters to prevent learning-based agents from making "obvious" mistakes, or modifying the reward function to more heavily penalize "trust-breaking" moves. Beyond the domain of Hanabi, other candidates for human-AI teaming experiments include StarCraft (which to date has only shown adversarial RL agent performance with humans [44]), Overcooked [9], and mixed cooperative-competitive environments like Bridge [49] and Diplomacy [16].

## 6 Conclusion

This study measured the game performance and human reactions in mixed human-AI teams in the cooperative card game Hanabi, comparing outcomes when humans were paired with a rule-based agent and when paired with a reinforcement-learning-based "Other-Play" agent designed to maximize zero-shot coordination. Despite achieving similar scores between these teams, human players strongly preferred working with the rule-based agent, and view the Other-Play agent quite negatively, citing their bilateral understanding, trust, comfort, and perceived performance as reasons. The ability of AI agents to team with humans is an important determinant of whether they can be deployed in many real-world situations. These results show that even state-of-the-art RL agents largely fail to convince humans that they are good teammates, and suggest that human perception of AI needs greater consideration in future AI design and development if it is to achieve real-world adoption.

## Acknowledgments and Disclosure of Funding

DISTRIBUTION STATEMENT A. Approved for public release. Distribution is unlimited. This material is based upon work supported by the Under Secretary of Defense for Research and Engineering under Air Force Contract No. FA8702-15-D-0001. Any opinions, findings, conclusions or recommendations expressed in this material are those of the author(s) and do not necessarily reflect the views of the Under Secretary of Defense for Research and Engineering. © 2021 Massachusetts Institute of Technology. Delivered to the U.S. Government with Unlimited Rights, as defined in DFARS Part 252.227-7013 or 7014 (Feb 2014). Notwithstanding any copyright notice, U.S. Government rights in this work are defined by DFARS 252.227-7013 or DFARS 252.227-7014 as detailed above. Use of this work other than as specifically authorized by the U.S. Government may violate any copyrights that exist in this work.

The authors would like to thank our experiment participants for their time. We thank Hengyuan Hu for providing the Other-Play model. We thank Peter Morales for his guidance during the early phase of this work.

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
