## Supplemental Materials

For data transparency and completeness, we detail our participant recruitment process, participant instructions, relevant aggregate data (demographics, scores, surveys), as well as the results from statistical tests that we conducted, but were not part of the main paper due to space constraints. We do not include the results of the NASA Task Load Index survey here because those were not analyzed for this study.

### 6.1  Participant Recruitment

Participants for this experiment were recruited via convenience and snowball sampling, with initial emails to MIT research groups and social mailing lists, as well as some for other Cambridge-areas groups. We note that 6 out of 29 participants belonged to the *hanab.live* Hanabi gaming community. Other than those, participants were novices to Hanabi, did not play consistently, or came from several distinct and unrelated Hanabi groups.

### 6.2  Introductory Slides and Game Interface

These are the slides shown to experiment participants at the very beginning of the session. All subjects were shown the same slides.

---

# Hanabi for Human-AI Teaming

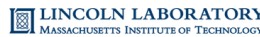

---

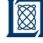  **Experiment Timeline**

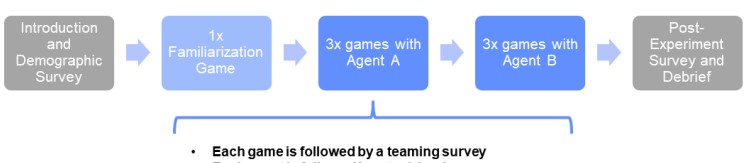

- Each game is followed by a teaming survey
- Each agent is followed by a task load survey

**You should stop the familiarization game when you feel comfortable with the interface.**

**You are free to stop the experiment and withdraw at any time.**

Human-AI Teaming 2
KCC 10/30/2020

LINCOLN LABORATORY
MASSACHUSETTS INSTITUTE OF TECHNOLOGY

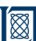

# Hanabi Rules and Game Interface

**You can see your teammate's cards, but not your own**
**You can see the hints you have given your teammate, and the hints they have given you**

The list of cards in the deck and both hands

The AI's cards

Hints you have given to the AI

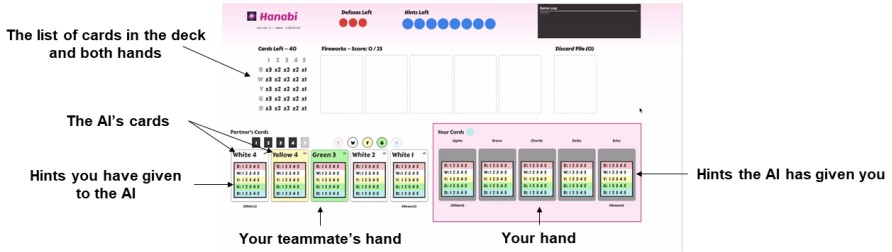

Hints the AI has given you

Your teammate's hand

Your hand

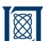

# Hanabi Rules and Game Interface

**On your turn you can….**

Hint a color

Hint a number

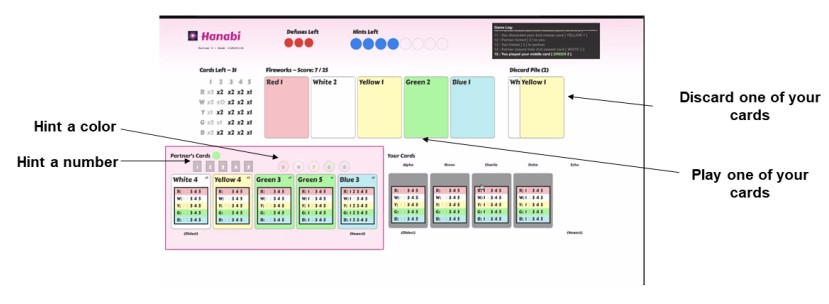

Discard one of your cards

Play one of your cards

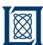

# Hanabi Rules and Game Interface

**The goal is to play 5 cards of each color in ascending order**
**i.e. Red 1, then Red 2, through Red 5**

If a played card is in the correct order it will be in this area

Moves are recorded here

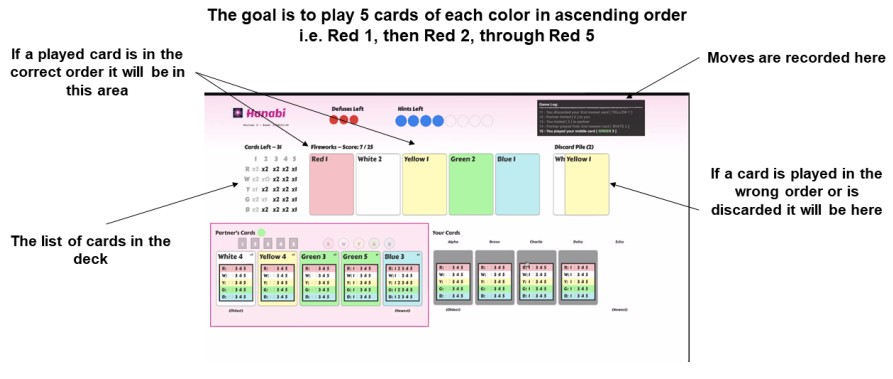

If a card is played in the wrong order or is discarded it will be here

The list of cards in the deck

# Hanabi Rules and Game Interface

**If a played card is in the wrong order, the defuses left will decrement**

**If a card is discarded, the hint tokens will increment**
**If a hint is given, the hint tokens will decrement**

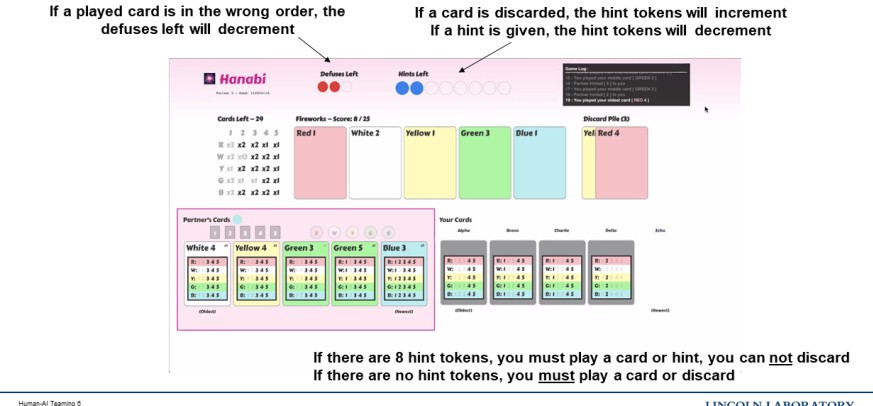

**If there are 8 hint tokens, you must play a card or hint, you can not discard**
**If there are no hint tokens, you must play a card or discard**

# Hanabi Rules and Game Interface

**Cards that have been played are greyed out**

**Cards that can still be played are black, and the number indicates how many are not in the discard pile**

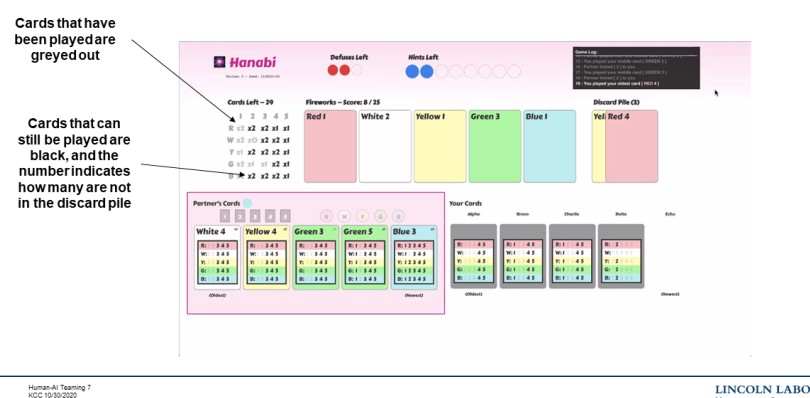

# Hanabi Rules and Game Interface

**The game ends under one of three conditions**

**There are no defuses left**

**After the last card is drawn, each player has one more turn**

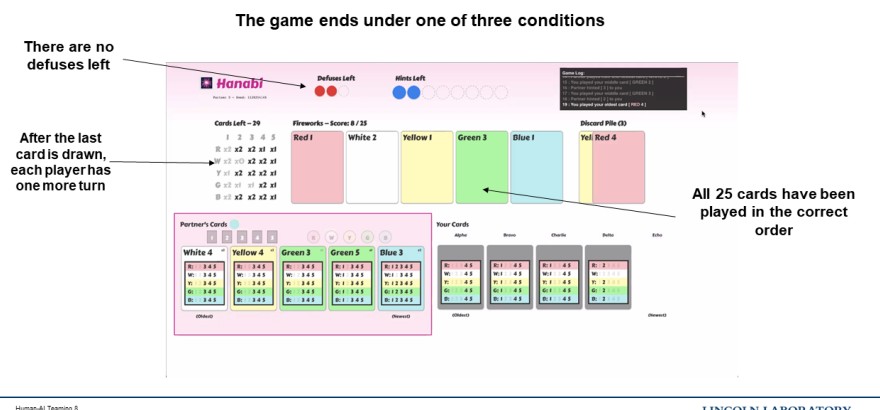

**All 25 cards have been played in the correct order**

## 6.3 Demographic Survey

Questions 1-4 are two pairs of multiple choice and free response questions. Questions 5-10 are Likert scale statements with a scale from 1 (strongly disagree) to 7 (strongly agree). Questions 9 and 10 also include a free response ("Explain") if the participant indicates agreement with the statement.

Table 4: Demographic Survey Prompt and Response Choices

|     | Prompt | Response Choices |
| --- | --- | --- |
| D1 | How often do you play card or board games? | [Never, <1 hour/week, 1-3 hours/week, >3 hours/week] |
| D2 | Which games or types of games do you play? | free response |
| D3 | How often do you play video games? | [Never, <1 hour/week, 1-3 hours/week, >3 hours/week] |
| D4 | Which game or types of video games do you play? | free response |
| D5 | I am experienced in cooperative card games. | Likert Scale |
| D6 | I am experienced in cooperative board games. | Likert Scale |
| D7 | I am experienced in cooperative video games. | Likert Scale |
| D8 | I am experienced in Hanabi. | Likert Scale |
| D9 | I am experienced in interacting with artificial intelligence agents (including voice assistants, game AIs, etc). | Likert Scale, free response (*optional*) |
| D10 | I am experienced in developing artificial intelligence agents. | Likert Scale, free response (*optional*) |

## 6.4 Demographic Survey Responses

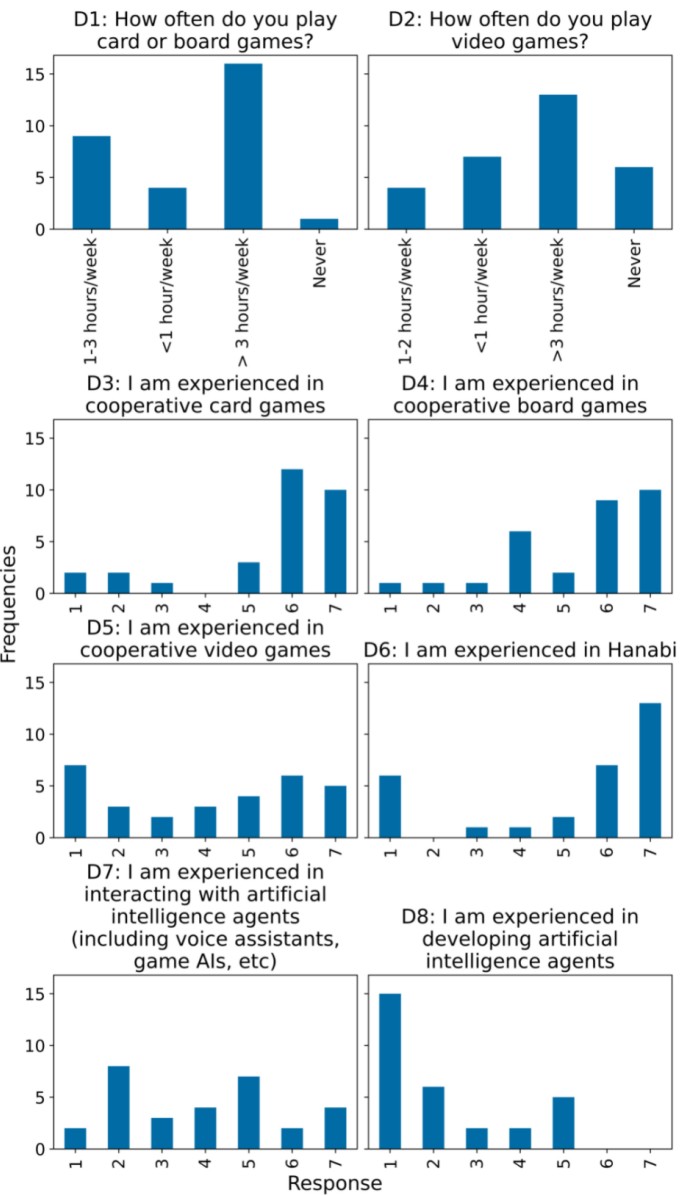

Figure 5: Histograms of all numerical and categorical demographic survey responses.

## 6.5 Post-Game Likert Scale Question Responses

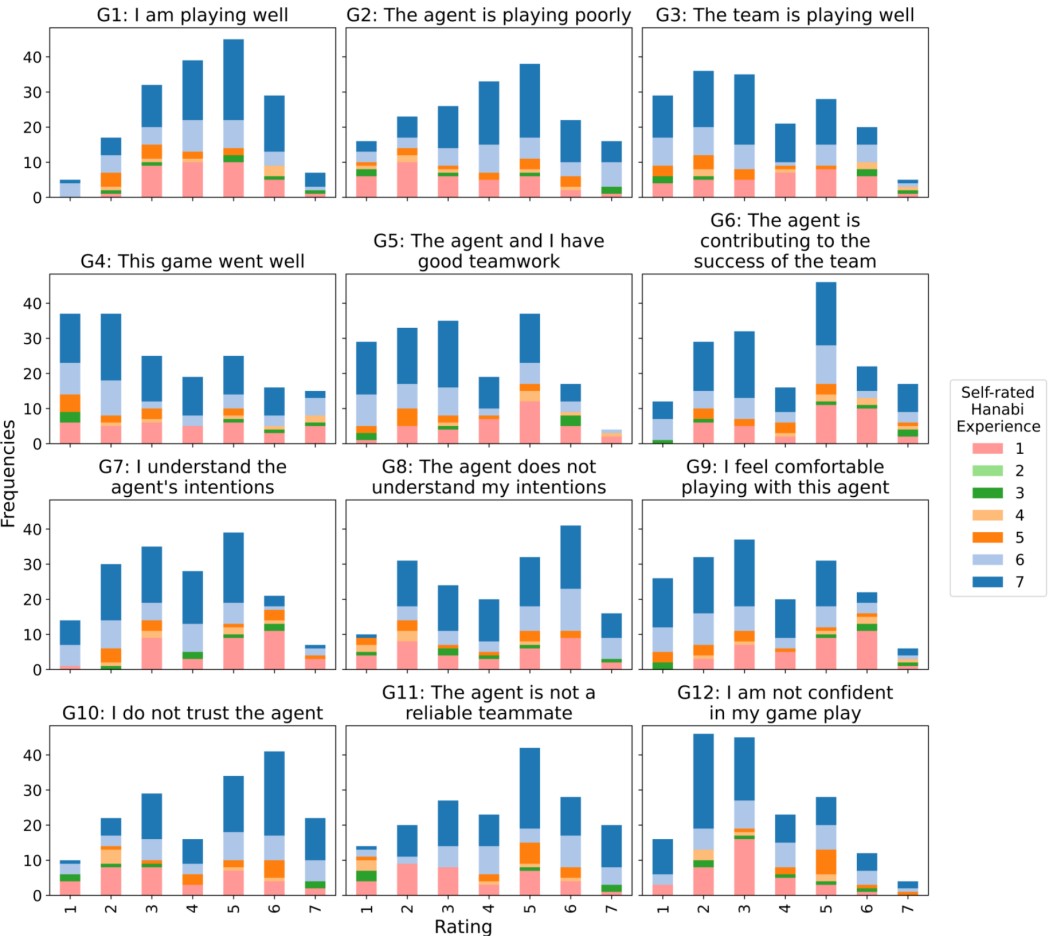

Figure 6: Participant rating for all post-game questions by self-rated Hanabi experience where statistically significant differences related to factors of agent and/or experience were presented in Section 4.2 . The scale ranges from 1-7, corresponding to "strongly disagree" to "strongly agree".

## 6.6 Participant Scores

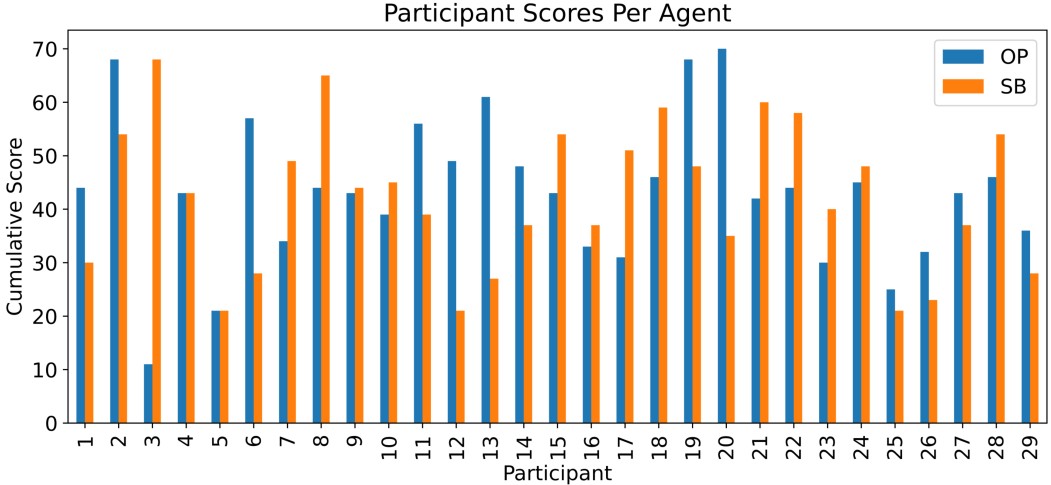

Figure 7: Cumulative game score for each participant across their six games, split by their three games with each agent type. The maximum achievable cumulative score per agent type is 75, and 150 for both. Participant 2 achieved the highest cumulative score of 122.

## 6.7 Post-Game Survey Statistics

Objective and subjective data were fit to second-order mixed-effects models with fixed factors of (1) AI agent, (2) self-rated Hanabi experience, (3) block (the first or second set of three games), and (4) game number (first, second, or third game within a block), and a random factor of participant number. AI agent and participant number were considered categorical variables.

*Agent* refers to the AI agent type (OP or SB).

*Experience* refers to the self-reported Hanabi experience level (question D8).

*Lower* and *Upper* are the values for the 95% confidence intervals.

Table 5: Model fit to response variable of game score

| Name | Estimate | SE | $t$ | df | $p$ | Lower | Upper |
|---|---|---|---|---|---|---|---|
| Agent | 1.6084 | 4.2962 | 0.37438 | 158 | 0.70863 | -6.877 | 10.094 |
| Block | 5.282 | 2.0039 | 2.6358 | 158 | **0.0092295** | 1.3241 | 9.2398 |
| Game | 2.9572 | 1.567 | 1.8872 | 158 | 0.060961 | -0.13765 | 6.0521 |
| Experience | 0.7736 | 0.62989 | 1.2281 | 158 | 0.22122 | -0.4705 | 2.0177 |
| Agent : Block | -3.6394 | 1.9493 | -1.867 | 158 | 0.063748 | -7.4893 | 0.21061 |
| Agent : Game | 1.1848 | 1.1962 | 0.99047 | 158 | 0.32346 | -1.1778 | 3.5473 |
| Block : Game | -0.90997 | 0.9963 | -0.91335 | 158 | 0.36245 | -2.8778 | 1.0578 |
| Agent : Experience | -0.20234 | 0.41707 | -0.48515 | 158 | 0.62824 | -1.0261 | 0.62141 |
| Block : Experience | 0.15829 | 0.36415 | 0.43468 | 158 | 0.66439 | -0.56094 | 0.87751 |
| Game : Experience | -0.13373 | 0.23366 | -0.57233 | 158 | 0.56791 | -0.59524 | 0.32777 |

Table 6: Model fit to response variable of "I am playing well" (G1)

| Name | Estimate | SE | $t$ | df | $p$ | Lower | Upper |
|------|----------|-----|-----|-----|-----|-------|-------|
| Agent | 1.9694 | 1.0546 | 1.8674 | 164 | 0.063637 | -0.11304 | 4.0518 |
| Block | 1.9168 | 0.39977 | 4.7948 | 164 | **3.6319e-06** | 1.1274 | 2.7062 |
| Game | 1.2204 | 0.30518 | 3.9989 | 164 | **9.6048e-05** | 0.6178 | 1.823 |
| Experience | 0.32729 | 0.12129 | 2.6985 | 164 | **0.0076942** | 0.087808 | 0.56678 |
| Agent : Block | -0.80197 | 0.62076 | -1.2919 | 164 | 0.19821 | -2.0277 | 0.42375 |
| Agent : Game | -0.37579 | 0.20773 | -1.809 | 164 | 0.072285 | -0.78597 | 0.034391 |
| Block : Game | -0.39684 | 0.18093 | -2.1934 | 164 | **0.029688** | -0.75409 | -0.039593 |
| Agent : Experience | -0.097033 | 0.073054 | -1.3282 | 164 | 0.18595 | -0.24128 | 0.047215 |
| Block : Experience | -0.12775 | 0.065443 | -1.952 | 164 | 0.052636 | -0.25696 | 0.0014721 |
| Game : Experience | -0.033548 | 0.041611 | -0.80624 | 164 | 0.42127 | -0.11571 | 0.048614 |

Table 7: Model fit to response variable of "The agent is playing poorly" (G2)

| Name | Estimate | SE | $t$ | df | $p$ | Lower | Upper |
|------|----------|-----|-----|-----|-----|-------|-------|
| Agent | -0.33608 | 1.0745 | -0.31277 | 164 | 0.75485 | -2.4577 | 1.7856 |
| Block | 0.83106 | 0.50514 | 1.6452 | 164 | 0.10184 | -0.16635 | 1.8285 |
| Game | 1.1238 | 0.3918 | 2.8684 | 164 | **0.0046687** | 0.35021 | 1.8975 |
| Experience | 0.62968 | 0.15572 | 4.0436 | 164 | **8.0826e-05** | 0.3222 | 0.93717 |
| Agent : Block | 0.45465 | 0.48308 | 0.94115 | 164 | 0.34801 | -0.49921 | 1.4085 |
| Agent : Game | -0.11362 | 0.29781 | -0.38151 | 164 | 0.70332 | -0.70165 | 0.47441 |
| Block : Game | -0.29889 | 0.24918 | -1.1995 | 164 | 0.23207 | -0.7909 | 0.19313 |
| Agent : Experience | 0.16086 | 0.10481 | 1.5348 | 164 | 0.12677 | -0.046092 | 0.36782 |
| Block : Experience | -0.16477 | 0.091074 | -1.8092 | 164 | 0.072251 | -0.3446 | 0.015057 |
| Game : Experience | -0.10097 | 0.058624 | -1.7223 | 164 | 0.086904 | -0.21672 | 0.014788 |

Table 8: Model fit to response variable of "The team is playing well" (G3)

| Name | Estimate | SE | $t$ | df | $p$ | Lower | Upper |
|------|----------|-----|-----|-----|-----|-------|-------|
| Agent | 1.3927 | 0.98213 | 1.418 | 164 | 0.15807 | -0.54654 | 3.3319 |
| Block | 2.4468 | 0.46171 | 5.2995 | 164 | **3.6985e-07** | 1.5352 | 3.3585 |
| Game | 1.065 | 0.35812 | 2.9739 | 164 | **0.0033848** | 0.35787 | 1.7721 |
| Experience | -0.032564 | 0.14234 | -0.22878 | 164 | 0.81933 | -0.31361 | 0.24848 |
| Agent : Block | -0.58469 | 0.44155 | -1.3242 | 164 | 0.18729 | -1.4565 | 0.28716 |
| Agent : Game | -0.22945 | 0.2722 | -0.84293 | 164 | 0.4005 | -0.76692 | 0.30803 |
| Block : Game | -0.56663 | 0.22776 | -2.4879 | 164 | **0.01385** | -1.0163 | -0.11692 |
| Agent : Experience | -0.19952 | 0.095802 | -2.0826 | 164 | **0.03884** | -0.38868 | -0.010353 |
| Block : Experience | -0.076342 | 0.083244 | -0.91709 | 164 | 0.36044 | -0.24071 | 0.088027 |
| Game : Experience | 0.051342 | 0.053584 | 0.95816 | 164 | 0.33939 | -0.054461 | 0.15715 |

Table 9: Model fit to response variable of "The game went well" (G4)

| Name | Estimate | SE | $t$ | df | $p$ | Lower | Upper |
|------|----------|-----|-----|-----|-----|-------|-------|
| Agent | 0.76539 | 1.1659 | 0.65651 | 164 | 0.51242 | -1.5366 | 3.0674 |
| Block | 2.0535 | 0.54808 | 3.7468 | 164 | **0.00024779** | 0.97132 | 3.1357 |
| Game | 1.8013 | 0.42511 | 4.2372 | 164 | **3.7659e-05** | 0.96187 | 2.6407 |
| Experience | -0.16132 | 0.16896 | -0.95475 | 164 | 0.34111 | -0.49494 | 0.17231 |
| Agent : Block | -0.34362 | 0.52415 | -0.65558 | 164 | 0.51301 | -1.3786 | 0.69133 |
| Agent : Game | -0.076741 | 0.32312 | -0.2375 | 164 | 0.81257 | -0.71476 | 0.56128 |
| Block : Game | -0.90768 | 0.27036 | -3.3573 | 164 | **0.00097855** | -1.4415 | -0.37384 |
| Agent : Experience | -0.17102 | 0.11372 | -1.5038 | 164 | 0.13455 | -0.39557 | 0.053529 |
| Block : Experience | 0.086306 | 0.098817 | 0.8734 | 164 | 0.38372 | -0.10881 | 0.28142 |
| Game : Experience | -0.00048774 | 0.063608 | -0.0076679 | 164 | 0.99389 | -0.12608 | 0.12511 |

Table 10: Model fit to response variable of "The agent and I have good teamwork" (G5)

| Name | Estimate | SE | t | df | p | Lower | Upper |
|------|----------|-----|---|-----|---|-------|-------|
| Agent | 2.3867 | 0.92567 | 2.5783 | 164 | **0.010807** | 0.55891 | 4.2145 |
| Block | 2.5072 | 0.43517 | 5.7614 | 164 | **4.0236e-08** | 1.6479 | 3.3664 |
| Game | 1.2232 | 0.33753 | 3.6238 | 164 | **0.00038684** | 0.55669 | 1.8896 |
| Experience | -0.026039 | 0.13415 | -0.1941 | 164 | 0.84634 | -0.29093 | 0.23885 |
| Agent : Block | -0.84574 | 0.41617 | -2.0322 | 164 | **0.043746** | -1.6675 | -0.024003 |
| Agent : Game | -0.36521 | 0.25656 | -1.4235 | 164 | 0.15649 | -0.87179 | 0.14137 |
| Block : Game | -0.59262 | 0.21467 | -2.7607 | 164 | **0.0064254** | -1.0165 | -0.16876 |
| Agent : Experience | -0.27275 | 0.090295 | -3.0207 | 164 | **0.0029261** | -0.45105 | -0.094464 |
| Block : Experience | -0.080736 | 0.078459 | -1.029 | 164 | 0.30499 | -0.23566 | 0.074185 |
| Game : Experience | 0.024633 | 0.050504 | 0.48775 | 164 | 0.62638 | -0.075089 | 0.12435 |

Table 11: Model fit to response variable of "The agent is contributing to the success of the team" (G6)

| Name | Estimate | SE | t | df | p | Lower | Upper |
|------|----------|-----|---|-----|---|-------|-------|
| Agent | 1.782 | 1.191 | 1.4963 | 164 | 0.13651 | -0.56962 | 4.1336 |
| Block | 2.441 | 0.48735 | 5.0086 | 164 | **1.4056e-06** | 1.4787 | 3.4032 |
| Game | 1.4781 | 0.37685 | 3.9222 | 164 | **0.00012878** | 0.73397 | 2.2222 |
| Experience | 0.1814 | 0.14897 | 1.2177 | 164 | 0.22509 | -0.11275 | 0.47556 |
| Agent : Block | -0.74292 | 0.65379 | -1.1363 | 164 | 0.25747 | -2.0338 | 0.548 |
| Agent : Game | -0.23663 | 0.2688 | -0.88034 | 164 | 0.37997 | -0.76739 | 0.29412 |
| Block : Game | -0.62886 | 0.23001 | -2.7341 | 164 | **0.0069428** | -1.083 | -0.1747 |
| Agent : Experience | -0.17078 | 0.094562 | -1.806 | 164 | 0.072747 | -0.3575 | 0.015934 |
| Block : Experience | -0.070518 | 0.083572 | -0.8438 | 164 | 0.40001 | -0.23553 | 0.094498 |
| Game : Experience | -0.065545 | 0.053426 | -1.2268 | 164 | 0.22164 | -0.17104 | 0.039946 |

Table 12: Model fit to response variable of "I understand the agent's intentions" (G7)

| Name | Estimate | SE | t | df | p | Lower | Upper |
|------|----------|-----|---|-----|---|-------|-------|
| Agent | 1.4938 | 0.86912 | 1.7188 | 164 | 0.087537 | -0.22226 | 3.2099 |
| Block | 2.6995 | 0.40858 | 6.6069 | 164 | **5.2348e-10** | 1.8927 | 3.5062 |
| Game | 1.6848 | 0.31691 | 5.3164 | 164 | **3.4175e-07** | 1.0591 | 2.3106 |
| Experience | -0.011884 | 0.12596 | -0.09435 | 164 | 0.92495 | -0.26059 | 0.23682 |
| Agent : Block | -0.55231 | 0.39074 | -1.4135 | 164 | 0.1594 | -1.3238 | 0.21922 |
| Agent : Game | -0.42332 | 0.24088 | -1.7574 | 164 | 0.08072 | -0.89895 | 0.05231 |
| Block : Game | -0.78483 | 0.20155 | -3.8939 | 164 | **0.00014328** | -1.1828 | -0.38686 |
| Agent : Experience | -0.19273 | 0.084778 | -2.2734 | 164 | **0.024301** | -0.36013 | -0.025335 |
| Block : Experience | -0.10565 | 0.073666 | -1.4341 | 164 | 0.15343 | -0.2511 | 0.039808 |
| Game : Experience | 0.019661 | 0.047418 | 0.41464 | 164 | 0.67895 | -0.073968 | 0.11329 |

Table 13: Model fit to response variable of "The agent does not understand my intentions" (G8)

| Name | Estimate | SE | t | df | p | Lower | Upper |
|------|----------|-----|---|-----|---|-------|-------|
| Agent | 1.1503 | 1.2751 | 0.90212 | 164 | 0.36831 | -1.3675 | 3.6682 |
| Block | 1.3412 | 0.51911 | 2.5838 | 164 | **0.010645** | 0.31625 | 2.3662 |
| Game | 1.3544 | 0.40121 | 3.3757 | 164 | **0.00091923** | 0.56216 | 2.1466 |
| Experience | 0.3328 | 0.15861 | 2.0982 | 164 | **0.037422** | 0.019612 | 0.64598 |
| Agent : Block | -0.84719 | 0.70374 | -1.2039 | 164 | 0.23038 | -2.2367 | 0.54236 |
| Agent : Game | -0.23748 | 0.28536 | -0.83219 | 164 | 0.40651 | -0.80093 | 0.32598 |
| Block : Game | -0.43934 | 0.24444 | -1.7974 | 164 | 0.074115 | -0.92199 | 0.043303 |
| Agent : Experience | 0.3178 | 0.10039 | 3.1657 | 164 | **0.0018446** | 0.11958 | 0.51602 |
| Block : Experience | -0.048055 | 0.088791 | -0.54122 | 164 | 0.58909 | -0.22338 | 0.12727 |
| Game : Experience | -0.10706 | 0.056744 | -1.8867 | 164 | 0.060972 | -0.2191 | 0.0049858 |

Table 14: Model fit to response variable of "I feel comfortable playing with this agent" (G9)

| Name | Estimate | SE | $t$ | df | $p$ | Lower | Upper |
|------|----------|-----|-----|-----|-----|-------|-------|
| Agent | 3.143 | 0.95194 | 3.3017 | 164 | **0.0011796** | 1.2633 | 5.0226 |
| Block | 2.5133 | 0.44752 | 5.616 | 164 | **8.1919e-08** | 1.6296 | 3.3969 |
| Game | 1.0899 | 0.34711 | 3.1398 | 164 | **0.0020055** | 0.40448 | 1.7752 |
| Experience | -0.023996 | 0.13796 | -0.17393 | 164 | 0.86213 | -0.29641 | 0.24841 |
| Agent : Block | -1.0492 | 0.42798 | -2.4516 | 164 | **0.015273** | -1.8943 | -0.20415 |
| Agent : Game | -0.34819 | 0.26384 | -1.3197 | 164 | 0.18877 | -0.86914 | 0.17277 |
| Block : Game | -0.49785 | 0.22076 | -2.2552 | 164 | **0.025444** | -0.93375 | -0.061956 |
| Agent : Experience | -0.33071 | 0.092857 | -3.5614 | 164 | **0.00048298** | -0.51406 | -0.14736 |
| Block : Experience | -0.081598 | 0.080686 | -1.0113 | 164 | 0.31336 | -0.24092 | 0.077719 |
| Game : Experience | 0.022646 | 0.051937 | 0.43602 | 164 | 0.66339 | -0.079906 | 0.1252 |

Table 15: Model fit to response variable of "I do not trust the agent" (G10)

| Name | Estimate | SE | $t$ | df | $p$ | Lower | Upper |
|------|----------|-----|-----|-----|-----|-------|-------|
| Agent | -0.80019 | 1.0903 | -0.73389 | 164 | 0.46406 | -2.9531 | 1.3527 |
| Block | 1.0878 | 0.51258 | 2.1221 | 164 | **0.03533** | 0.075645 | 2.0999 |
| Game | 1.1899 | 0.39758 | 2.9929 | 164 | **0.0031911** | 0.40487 | 1.9749 |
| Experience | 0.6055 | 0.15802 | 3.8318 | 164 | **0.00018089** | 0.29349 | 0.91752 |
| Agent : Block | 0.83804 | 0.4902 | 1.7096 | 164 | 0.089233 | -0.12987 | 1.806 |
| Agent : Game | -0.20681 | 0.3022 | -0.68436 | 164 | 0.49472 | -0.80351 | 0.38989 |
| Block : Game | -0.37637 | 0.25285 | -1.4885 | 164 | 0.13854 | -0.87564 | 0.1229 |
| Agent : Experience | 0.203 | 0.10636 | 1.9086 | 164 | 0.058055 | -0.0070082 | 0.413 |
| Block : Experience | -0.17691 | 0.092417 | -1.9143 | 164 | 0.057328 | -0.35939 | 0.0055706 |
| Game : Experience | -0.076328 | 0.059488 | -1.2831 | 164 | 0.20127 | -0.19379 | 0.041133 |

Table 16: Model fit to response variable of "The agent is not a reliable teammate" (G11)

| Name | Estimate | SE | $t$ | df | $p$ | Lower | Upper |
|------|----------|-----|-----|-----|-----|-------|-------|
| Agent | 0.34397 | 1.1345 | 0.3032 | 164 | 0.76212 | -1.8961 | 2.584 |
| Block | 1.0037 | 0.53332 | 1.8821 | 164 | 0.061598 | -0.049315 | 2.0568 |
| Game | 1.0982 | 0.41366 | 2.6549 | 164 | **0.0087148** | 0.28145 | 1.915 |
| Experience | 0.51945 | 0.16441 | 3.1594 | 164 | **0.0018826** | 0.19481 | 0.84409 |
| Agent : Block | 0.12847 | 0.51003 | 0.25188 | 164 | 0.80145 | -0.87861 | 1.1355 |
| Agent : Game | -0.1499 | 0.31442 | -0.47675 | 164 | 0.63417 | -0.77074 | 0.47094 |
| Block : Game | -0.30352 | 0.26308 | -1.1537 | 164 | 0.2503 | -0.82299 | 0.21595 |
| Agent : Experience | 0.12813 | 0.11066 | 1.1579 | 164 | 0.24859 | -0.090369 | 0.34664 |
| Block : Experience | -0.088578 | 0.096156 | -0.9212 | 164 | 0.3583 | -0.27844 | 0.10128 |
| Game : Experience | -0.097645 | 0.061895 | -1.5776 | 164 | 0.11659 | -0.21986 | 0.024569 |

Table 17: Model fit to response variable of "I am not confident in my gameplay" (G12)

| Name | Estimate | SE | $t$ | df | $p$ | Lower | Upper |
|------|----------|-----|-----|-----|-----|-------|-------|
| Agent | 1.4796 | 1.2583 | 1.1758 | 164 | 0.24137 | -1.005 | 3.9641 |
| Block | 1.5481 | 0.4575 | 3.3839 | 164 | **0.00089398** | 0.64478 | 2.4515 |
| Game | 0.61849 | 0.34399 | 1.798 | 164 | 0.074017 | -0.060728 | 1.2977 |
| Experience | 0.26304 | 0.13801 | 1.906 | 164 | 0.058405 | -0.0094638 | 0.53554 |
| Agent : Block | -1.1132 | 0.76425 | -1.4565 | 164 | 0.14716 | -2.6222 | 0.39588 |
| Agent : Game | 0.037263 | 0.22698 | 0.16417 | 164 | 0.8698 | -0.41092 | 0.48544 |
| Block : Game | -0.29982 | 0.20014 | -1.498 | 164 | 0.13605 | -0.69501 | 0.095368 |
| Agent : Experience | 0.031346 | 0.079802 | 0.39279 | 164 | 0.69498 | -0.12623 | 0.18892 |
| Block : Experience | -0.044646 | 0.07217 | -0.61861 | 164 | 0.53703 | -0.18715 | 0.097857 |
| Game : Experience | -0.048831 | 0.045717 | -1.0681 | 164 | 0.28704 | -0.1391 | 0.041439 |

## 6.8 Novice vs Expert Post-Game $t$-Tests

Post-hoc pairwise comparisons of novice vs expert in cases where agent and self-reported Hanabi experience have significant interaction effects, as described in Section 4.2.

Table 18: Two-sample $t$-tests of post-game sentiment, comparing novice and expert reactions.

| Question | Agent | $t$ | $p$ | corrected $p$ | $d$ |
|---|---|---|---|---|---|
| G5 | SB | 0.35599 | 0.72273 | 1.00000 | 0.080708 |
| G5 | OP | 5.1395 | 1.7334e-06 | **1.38672e-05** | 1.0185 |
| G9 | SB | 0.25536 | 0.79906 | 1.00000 | 0.057915 |
| G9 | OP | 5.8552 | 8.7246e-08 | **7.85214e-07** | 1.1214 |
| G8 | SB | 0.61126 | 0.54266 | 1.00000 | 0.13838 |
| G8 | OP | -5.9229 | 6.5231e-08 | **6.52310e-07** | -1.1306 |
| G3 | SB | -1.1856 | 0.2391 | 0.956400 | -0.26679 |
| G3 | OP | 3.5514 | 0.00062779 | **3.76674e-03** | 0.75189 |
| G7 | SB | 1.652 | 0.10223 | 0.511150 | 0.36893 |
| G7 | OP | 5.0678 | 2.3171e-06 | **1.62197e-05** | 1.0076 |

## 6.9 Post-Experiment $t$-Tests

One-sample $t$-tests of post-experiment sentiment. Some responses were flipped on the Likert scale for directional consistency, based on which agent was seen first, since the ends of the scale were labeled as the "first" and "second" agent for the participants. Preference directionality is such that 1 is towards OP and 7 is towards SB. $t$ statistics greater than 0 indicate answering towards SB. The Holm–Bonferroni step-down method was used for multiple comparisons correction. $d$ is the Cohen's effect size. In general, thresholds for "small," "medium," and "large" effect sizes are considered to be $|d| = 0.2, 0.5,$ and $0.8$ respectively [10].

Table 19: One-sample $t$-tests of post-experiment sentiment.

| Question | $t$ | $p$ | corrected $p$ | $d$ |
|---|---|---|---|---|
| Which agent did you prefer playing with? | 2.90633 | 0.00707 | **0.03969** | 0.549 |
| Which agent did you trust more? | 3.40564 | 0.00201 | **0.01610** | 0.644 |
| Which agent did you understand more? | 2.88618 | 0.00743 | **0.03969** | 0.545 |
| Which agent understood you better? | 2.93369 | 0.00661 | **0.03969** | 0.554 |
| Which agent was the better Hanabi player? | 3.36011 | 0.00226 | **0.01610** | 0.635 |
| Which agent was more reliable? | 2.86217 | 0.00788 | **0.03969** | 0.541 |
| Which agent had a better understanding of the game on average? | 2.68186 | 0.01214 | **0.03969** | 0.507 |
| Which agent caused you to have a greater mental workload? | -0.16385 | 0.87103 | 0.87103 | -0.031 |

## 6.10 Post-Experiment Participant Preference and Free Response

Post-experiment ratings of agent preference and explanation of the preference. The "Preference" heading corresponds to a Likert-scale response to the question "Which agent did you prefer playing with?" where 1 was "the first agent" and 7 was "the second agent." "Explanation" was a free-response field with the question "Why did you prefer the agent that you did?"

Table 20: Post-experiment ratings of agent preference and explanation

| Participant | Order | Preference | Explanation |
|---|---|---|---|
| 1 | SB, OP | 5 | first agent felt like it was learning; really bad to begin with; had to "teach" them how to play hanabi; second agent felt like someone who knew how to play hanabi and wanted to trick you; broke my trust in 2nd game; in 3rd game was "trust me"; don't like playing with 2nd agent. |

| 2 | OP, SB | 6 | It seemed to have a better understanding of not just what hints to give, but when to give them. I think a lot of the strategies cascaded from that - both my strategy and its. It just had a better understanding of the tempo of the game. If you think of it as - every time I get a hint, I have to perform an MLE, and when I give a hint, that's what they have to perform - if you just take the clue at face value; you can think about "why am I giving this hint" the second agent thought about "why am I getting/giving this hint NOW" while the first agent didn't. |
|---|---|---|---|
| 3 | SB, OP | 1 | It knew the rules of the game. It knew how to play. |
| 4 | OP, SB | 4 | I thought the first one was dumber but more consistent. The second one - I thought I was starting to understand it in the second game, but then in the third game, I completely didn't understand what it was doing at all. |
| 5 | SB, OP | 1 | Second agent made an obvious mistake quite frequently. There are some cases where it was clear that if the agent played a card, we would lose the game, but it played it anyway. Sometimes it would also give me hints that I already know. |
| 6 | OP, SB | 7 | better able to understand what it's clues meant and how to give it clues that would result in the correct actions |
| 7 | SB, OP | 3 | The first agent was more predictable, even if I didn't necessarily agree with their strategy. Both of them made dumb choices, like playing cards that were clearly not playable when they had full information on them (or at least enough information), or they discarded cards with full information and were playable. |
| 8 | OP, SB | 7 | The second agent seemed to be more capable of inductive reasoning than the first. Both has similar styles of inductive clues, but it seemed like the second took inductive clues better. The discard strategy of the first agent felt worse than the discard strategy of the second. |
| 9 | SB, OP | 1 | agent 1 was more consistent; even if i didn't understand what they were doing, i could more reliably assume they would play or discard cards if they knew they were playable; I feel I bombed the second one whenever i clued it; did not know how it would react |
| 10 | OP, SB | 6 | Maybe it's because the first one was so terrible that it made me have zero expectation of the second one. So even though the second agent wasn't that much better, and I was confused by its strategy, I was used to being confused and wasn't surprised anymore. It took less willpower to go through the games [with the second agent]. |
| 11 | SB, OP | 7 | Gave me info; seemed to act on cues better; it felt like there was 2-way comms as opposed to 1-way; also it didn't throw away cards (e.g., knew perfect info on) |
| 12 | OP, SB | 7 | It provided more challenge and interest. Because I could reasonably play with it. It let me play at a more satisfying level. |

| 13 | SB, OP | 1 | Agent 1 seemed to have a better model of the game in the sense that it deliberately played playable cards and discarded unplayable cards more frequently; as opposed to the 2nd agent that played known unplayable cards and did not play cards when it had the chance to; first agent was more inline with game of Hanabi rules of playing all cards when possible; first agent played a way that was more familiar; First agent still used strategies that were more human friendly; i understood it better and it understood me better |
|---|---|---|---|
| 14 | OP, SB | 3 | I felt like the first agent was improving and started understanding my strategy more, whereas the second one wasn't learning from the errors or mistakes that both of us made. |
| 15 | SB, OP | 6 | The rules that the second (2nd) agent was following was easier to understand; specifically the discarding strategy was much more predictable; first agent may have predictable discarding strategy, but the 2nd agent is much easier to play with |
| 16 | OP, SB | 7 | I probably had some learning effects for the game so I understood things better, however, I also found that it was easier to get into a cadence of play with the second agent. I think I understood the intention of the second agent and it understood me. |
| 17 | SB, OP | 1 | The first agent played cards and hinted cards consistently. The second agent by contrast did not play multiply-hinted cards and gave hints that were not necessarily playable. With the first agent, I could reasonably expect to perform well and to trust his decision whereas with the second agent, I found myself trying to compensate for his lack of reliability. |
| 18 | OP, SB | 2 | i was better at predicting what the first agent would do; after first 2 games i understood the agent's strategy though i didn't agree with it; with 2nd agent i couldn't figure out how it's saves and discards worked and that made it impossible for me to tell it to save cards i wanted to protect |
| 19 | SB, OP | 1 | because i could understand what it would do and i can predict what they would do better; and i have opinion that i can understand the clue of the first agent and what the agent tries to force me to do; the first agent preferred to play instead of discarding; second agent prefers to discard instead of play which is sub-optimal in hanabi game (i.e., it had full info about a card and still chose to discard) |
| 20 | OP, SB | 7 | It does understand rules of Hanabi among humans. |
| 21 | SB, OP | 7 | It gave me more hints and it didn't make inexplicable discard decisions that were clearly suboptimal based on information that it had at the time. It was also the only one of the six games that we completed (25 pts). |
| 22 | OP, SB | 6 | second agent played color clues that i gave |
| 23 | SB, OP | 5 | I felt that the second agent understood some clues better than the first even though i think they are very very similar; similar strategy on saving discarding cluing; main difference for the second one was that it would clue sooner than the first one; it wouldn't delay cluing even though it had cards; 3 games is a bit short to determine/assess strategy; |
| 24 | OP, SB | 6 | The main reason was that Agent 2 was willing to change its discard behavior to match mine, as I strongly prefer discarding the oldest card instead of the newest. The other reason is the second agent was a little better at giving clues to me that I understood the meaning of. |

| 25 | SB, OP | 7 | It seemed more cooperative in that it was giving a lot of hints and it seemed like we had a similar strategy. Early on, we'd tell each other when we had ones, and then giving full information, giving the appropriate hints for the state of the game. Seemed like we had good teamwork. They were giving hints, and also taking hints. |
|----|--------|---|---|
| 26 | OP, SB | 7 | The second agent understood my strategy better; it was easier for me to follow it's pattern/strategy; and because we got closer to winning, ergo, it was doing something right |
| 27 | SB, OP | 2 | the first agent provides more certainty even though the game progresses slower, it acts upon certainty and minimizes guessing; |
| 28 | OP, SB | 6 | i felt like the second agent was playing with easier to understand set of rules; they appeared to be more mindful of hints or number of hints remaining, so there is a better back-and-forth depending on who what playable cards or not; |
| 29 | SB, OP | 3 | To my understanding, the strategy seemed very consistent and simple. Agent 2's strategy seemed more complex and less predictable. It seemed more random which is less preferable. |