# OpenReview forum: "Evaluation of Human-AI Teams for Learned and Rule-Based Agents in Hanabi"
_NeurIPS.cc/2021/Conference — NeurIPS 2021 Poster_

### Official Review · Reviewer_mR2Y · 2021-07-07

**Rating:** 5
**Confidence:** 3

**Summary:**

This paper experimentally evaluates player perception and subjective experience when playing Hanabi in a team with a learning or rule-based agent. A quantitative analysis of player survey responses indicates that while human-agent teams performed similarly in terms of score for either type of agent, players preferred to play with the rule-based agent.

The paper is very well written, and Hanabi is particularly well explained. The subject of the paper, and its results, are also quite interesting. My main criticism of the paper is that there is not enough qualitative analysis of player responses. The quantitative analysis shows a preference for the rule-based agent, and I can make assumptions as to why, but I wanted to read some analysis of the likely reasons based on player responses. I think this is important as this kind of study should give us some insight into what human players expect or desire in an agent teammate. I assume that player feedback was elicited as some responses are discussed for the survey question on mental workload.


**Limitations And Societal Impact:**

Yes?

**Main Review:**

Page 6: How should we interpret the games scores by self-rate player experience plot in Figure 1? When I look at it, I see a correlation between player score and self-rated experience for OP games, but then the commentary says the correlation is not significant for OP games. Could the authors elaborate on this more, and provide more commentary for Figure 1? Given that the result for OP games -- that there is no significant correlation between self-rated player experience and score -- is a bit counter-intuitive, there needs to be more discussion here as to the likely reasons why this is the case. Why do you think there was correlation for SB games and not OP?

In the results where self-rated Hanabi experience is referred to, I would add the "self-rated" prefix so that a reader doesn't confuse "Hanabi experience" with "subjective game-play experience" (i.e., how they rated their experience in playing with a particular agent). I did this on my first read through the paper.

I did find the use of OP and SB to denote the two agents used in the experiments confusing sometimes, as I had to keep reminding myself that OP was the RL agent and SB was the rule-based agent. The acronyms do make sense given the agents they are referring to, but it would have made my life easier if they were something like LB-OP and RB-SB. This is a minor point, however.


**Time Spent Reviewing:**

3

---

> ### Author Response · Authors · 2021-08-10
> **initial response to reviewer mR2Y**
>
> Thank you for the review and the suggested changes.
>
> Regarding qualitative analysis subject comments – we do indeed have many subject comments recorded, both as part of open-ended survey responses and as part of experimenter observation notes. There was simply not enough space in this paper to discuss these items in detail (we are at the page limit). Although we mention some of the player responses briefly (242-244), we are planning follow-up work to go deeper into these results. Furthermore, we can add the subject commentary/responses to the supplementary material for completeness.
>
> We would interpret the significant correlation of player experience with performance in the rule-based agent but not the learning-agent to mean that experienced Hanabi players are unable to leverage their previous experience to perform notably better than novices when working with the OP learning-based agent. We can clarify this in text and tie it into the discussion of novice vs expert reactions as well.
>
> Thank you for the suggestion on adding “self-rated” to “Hanabi experience” for clarity. We will make that edit.
>
> Once again, thank you for the time and effort for this review.

---

### Official Review · Reviewer_zPDw · 2021-07-10

**Rating:** 6
**Confidence:** 5

**Summary:**

The paper describes an experiment where humans were playing against bots in the game of Hanabi, a cooperative card game. There were two types of bots: rule based, and reinforcement learning based. Even though the two types played equally strong, the humans, when asked, strongly preferred the rule based bots.
This result holds implications for design of future human-computer interaction bots.

**Ethical Concerns:**

There appear to be no ethical concerns. Human subjects have given their consent.

**Limitations And Societal Impact:**

Ethical and limitations are dealt with in an appropriate manner in this paper. The limitations are used to show future work.

**Main Review:**

The paper is very well written. The English is perfect, the text is very clear.
The originality of the paper is high: this is a social science experiment with 29 human participants to see if they like a rule based bot or a reinforcement learning based bot. No algorithms were developed for this paper by the authors. The outcome may be surprising, and is certainly interesting.
The qulaity of the work is good, the statistical analysis is elaborate, and enough details are provided about setup and analysis.
The paper is significant, as so far as results based on a single card game can be considered significant. The results are certainly refreshing.


**Time Spent Reviewing:**

2

---

> ### Author Response · Authors · 2021-08-10
> **initial response to reviewer zPDw**
>
> Thank you for your kind review!

---

### Official Review · Reviewer_aZxX · 2021-07-20

**Rating:** 5
**Confidence:** 4

**Summary:**

This paper examines human-AI collaboration (“teaming”) in Hanabi. It seeks to expand current research on Hanabi in two ways: (1) comparing performance of a rule-based policy and a learned policy (for the AI teammate), and (2) measuring agent performance through both objective and subjective metrics. The two policies achieved comparable performance in games with human participants; however, participants tended to subjectively evaluate the rule-based policy more positively than the learned policy.

**Limitations And Societal Impact:**

I would encourage the authors to strengthen their discussion of two limitations. First, their findings are specific to these agents -- without further empirical or conceptual evidence, the results can’t be immediately translated to rule-based and learned policies in general. Second, I would broaden the disavowal of generalization beyond Hanabi. The “tight coupling of coordination and performance, and the codification of communication” are not the only reasons to hesitate extending these conclusions to other domains. Would the authors expect these findings to generalize to a spatially extended game like Overcooked? What about a domain in which rule-based policies and learned policies did not perform comparably (vis a vis the lack of correlation between objective performance and subjective preferences)?

Otherwise, I thought the authors adequately addressed limitations of their work. (They did not address negative societal impact.)

**Main Review:**

*Originality.* This paper helps pioneer an important topic of research for both human-AI collaboration and Hanabi researchers: subjective perceptions of and preferences over agent partners (in the context of human-AI co-play). As the authors indicate, relatively few Hanabi studies have tested how agent performance generalizes to human partners. Critically, though the paper does not develop any new algorithms or improve upon existing algorithms, it breaks new ground by drawing attention to the importance of human preferences and pioneering the subjective measurement of agent effectiveness. Related work is adequately cited.

*Quality.* I think the submission has the potential to be a solid paper, but currently its quality falls into an intermediate zone of sorts. Four areas for improvement occur to me as I read the current draft (and could revise my rating if addressed):

1) Disclarity in the paper’s background. The information communicated in Section 2.2 is central to the design decisions underlying the paper, but in practice I found the explanation(s) muddled. In particular...

   - The discussion of self play from lines 67-73 seems overly convoluted. Isn’t it straightforward to say (and understand) self play refers to an agent playing with a copy of itself, either during training or for an evaluation? The training-evaluation distinction is frequently made in RL research, so it seems unnecessary to spend (limited) space explaining that self play can be used in two “closely related but [...] not synonymous” contexts. (Also, in contrast to the expectations set by lines 72-73, I didn’t see any need for disambiguation throughout the paper.)
    - Does “human play” refer to any evaluation-time play with humans (e.g., even if human play was part of the training loop), or specifically evaluation-time play with humans not encountered during training? If the former, human play should be defined in a paragraph separate from the cross-play paragraph. If the latter, I would recommend clarifying by changing “referred to as human-play” to “which we refer to as human-play”.
    - The paragraph from lines 81-87 is very difficult to follow. What are the core characteristics that define zero-shot coordination and ad-hoc teaming, and how are they mutually exclusive? It’s absolutely fine to choose one paradigm over the other for the experiments at hand, but I was left wondering whether there are any practical differences (especially given the parenthetical statement on lines 87-87).

- Section 2.3 presented similar questions:
    - What were the criteria used to select the studies included in Table 1? Section 2.3 offers a surface-level description (“most salient”), but doesn’t clarify what metrics (or prior paper) gives this selection weight. (I think it's also fine to just say "We surveyed the literature and analyzed agent performance across the cited papers".)
    - What does “exhaustive numerical comparison” (in the Table 1 notes) mean? The idea that we shouldn’t draw any inferences from these numbers is in tension with the bolding of certain (high) values in the table. To add to the confusion, two cross-play values that are separated by 0.5 are both bolded, whereas a self-play value within 0.3 of the top self-play score for rule-based agents is not bolded.
    - What are the actual defining characteristics for “rule-based” and “learning-based” agents? Lines 91-93 offer “tend” definitions (“Rule-based agents tend…”), without clarifying what defines or differentiates these two categories.
 - In Section 2.4, many of the “metrics” introduced appear to be concepts rather than metrics. Indeed, they are never operationalized or connected to the actual variables measured in Section 3.1 (and Table 2). How do these map to the measures chosen for the study?
 - Finally, lines 168-172 appear to concern findings from a prior, related study, and should likely be relocated to the background section.

2) Lack of method transparency. The authors take several steps to support transparency and reproducibility (e.g., including the tutorial materials and demographic questionnaire in their supplementary files). However, there are several critical pieces missing for method transparency, particularly from a behavioral research perspective.

    - Where were participants recruited from? The generalizability of findings from the paper will be different depending on participant representativeness (which differs between sources like Mechanical Turk or department colleagues).
    - How much information were participants provided about the experiment? Were participants told they would play with two different players? Were they informed of the pay structure in advance? (This relates to how strongly participant performance was incentivized; i.e., whether they cared at all how they performed.)
    - How were the statements for the Likert-type scales chosen? In the paragraph on lines 155-158, it would be helpful if the authors either cited prior studies or stated their intention with selecting or generating these scale items (e.g., “To measure the perceived legibility of the agent teammates, …”).
    - What part of the experiment was administered by video conference? Many (if not most) online experiments are conducted without need for experimenter intervention via video call, so this was a bit unexpected.
    - Where does the verbal participant commentary described on lines 240-244 come from, in terms of the experimental procedure? It doesn’t appear to be described anywhere in Section 3.1.

3) Adequate (but not stellar) analysis. I agree with the authors’ overall conclusion -- that objective performance does not differ with the two agents, but subjective preferences do -- based on the analysis that they present. However, there is a good amount of room to improve their analysis and interpretative work.

    - The factorial study design (2 [agent type] x 2 [game block]) indicates that an interaction term should be included between agent type and game block within the mixed-effect models. When interaction terms are excluded in these situations, it is possible for underlying interaction effects to incorrectly register as main effects (even if potentially unlikely with the authors’ current findings).
    - Given the large number of Likert items completed by participants, I was surprised the authors did not try to generate any combined measures. For example, Likert scales (properly defined) are the result of summing responses for multiple, *a priori* selected items. Another option would be to conduct factor analysis and use the first *N* factors. Either approach reduces the total number of statistical tests run in a way that accounts for the potential correlation between participants’ responses. These approaches may also encourage us to think about the underlying constructs we are attempting to study (e.g., legibility or predictability).
    - I would discourage the authors from discussing “near-significance” (lines 203-204). Adding a new interpretative threshold for marginal significance inflates false positive report probabilities. (I also notice that a *p*-value of 0.061 is interpreted as “near-significant” but elsewhere a value of 0.0867 is "non-significant").
    - I encourage the authors to report full results regression tables in their supplement for better results transparency and reproducibility.
    - (As one final incidental comment on the results, doesn’t the paragraph on lines 222-225 say the exact same thing as the paragraph on lines 226-233? Does having both paragraphs add any particular value? It took me a read or two to realize they might be discussing the same finding.)

4) (Slightly) overstretched discussion claims. In general, I enjoyed the writing and argumentation in this paper. However, I feel that there were a few small areas in the discussion and conclusion where the evidence at hand is stretched a bit further than is plausible. The paper makes several claims about human approaches to Hanabi (“Human Hanabi players typically do not learn with this assumption”, “common practices”, “human learning [is] more akin to few-shot coordination”, lines 250-253) without citing any sources or offering evidence. The conclusion also claims that participants “cit[e] their bilateral understanding, trust, comfort, and perceived performance as reasons” (lines 307-308). Was this substantiated at any point in the paper (or in the data gathered in the experiment)? As I understand it, the design could offer at best correlational support for a link between preferences and perceptions of performance, etc.

*Clarity.* The paper is mostly well written, though the “unclear explanations” and “lack of method transparency” concerns elaborated previously impinge on clarity.

*Significance.* Multiple research teams have noted that Hanabi presents an interesting challenge domain for AI research. This paper substantially advances the state of knowledge around Hanabi agents, given the unique research approach, data collected, and questions posed. Though the authors note that the costs of human evaluation are high relative to AI research, I hope this paper will encourage others to engage in more human-AI collaboration research (and within that space, collect data on human perceptions and preferences).

**Time Spent Reviewing:**

7

---

> ### Author Response · Authors · 2021-08-10
> **initial response to reviewer aZxX**
>
> Thank you for the review and the suggested changes.
>
> On clarity: We agree with most all of the suggestions and will work to adopt them in a final draft.  We agree that the definition of self-play can be simplified by adopting the reviewer’s recommendation. Human-play only referred to evaluation-time; no literature was found that involved human-in-the-loop training; the only human elements within the training loop were policies that learned from human data via behavior cloning. Zero-shot coordination refers to training that aims to maximize performance with a previously-unseen teammate, while ad-hoc teaming focuses on an agent’s ability to update its policy during interaction with a teammate, and the teammate may come from a previously-known set. The two are not mutually-exclusive – an agent trained for high zero-shot performance may also then use ad-hoc teaming to improve performance over time – though the objectives and problem setup differ. We can clarify the paragraph from 81-87.
>
> Table 1 included all of the distinct Hanabi AI that we found in the academic literature with at least self-play scores. “Exhaustive numerical comparison” could perhaps be better phrased. We meant that the evaluation conditions for each column are not the same per agent, particularly for cross-play and human play, so we would urge some amount of caution when comparing the scores. This is described parenthetically, but we can rephrase to avoid confusion. There is no way to put them on a level playing field for comparison without effectively re-performing all of the cited experiments, so for the purposes of highlighting particularly high performers, we do think that bolding is appropriate (though we agree there is some tension there). The 13.3 that was bolded on cross-play was likely a mistake on our part, and we can remove the bolding.
>
> Thank you for the suggestion on lines 91-93 for rule-based and learning-based agents. Perhaps the following might be better? _“Rule-based agents have a decision policy composed of a predefined set of rules to follow given any particular game situation, and the rules are often derived from human domain knowledge. Learning-based agents, on the other hand, use  statistical learning methods to adjust the parameters of their policy. The mechanism governing what action the policy will choose is learned via the agent’s experience, without the need for human domain knowledge.”_
>
> On Section 2.4 – it is a fair point that the subjective “metrics” and the concepts described in 123-134 are better termed “constructs” in the human factors literature, and we could indeed tie them better to the Likert scale statements. We will make that adjustment to the text, and add a key to Table 2 to tie the statements back to constructs.
>
> We can shift lines 168-172 about the Liang and Hu papers to the previous subsection in the Background.
>
> The reviewer presents an excellent set of clarifying questions to enhance experiment transparency and reproducibility. We will take these items into consideration and revise the supplementary material accordingly, but to address them here (questions shorted for character limit):
>
> ### _Where were participants recruited from? The generalizability…_
>
> The participants came from a fairly diverse pool, since many (but not all) were referred to us via word-of-mouth among several distinct and unrelated groups of Hanabi players (as well as individuals who did not regularly play Hanabi, or play Hanabi at all). Participants included students at several universities (affiliated to the researchers and not), unaffiliated researchers, unaffiliated working professionals, and several individuals from an online Hanabi community. Mechanical Turk was not used, and there was not a majority of recruitment from any single source.
>
> ### _How much information were participants provided about the experiment? Were participants told…_
>
> Participants were told that they would be playing with two different AI teammates, and they were informed of the pay structure in advance, but no subjects mentioned any tie between their preferences/behavior and the pay structure during or after the experiment, while many indicated that they participated purely out of interest. They were not told of the nature or names of the AI agents, but we did explicitly tell them they were playing with two distinct.
>
> ### _How were the statements for the Likert-type scales chosen? In the paragraph on..._
>
> We will add the related constructs that the questions were intended to address in Table 2, per above. The statements were largely derived from a compilation in Hoffman, Guy. "Evaluating fluency in human–robot collaboration." IEEE Transactions on Human-Machine Systems 49.3 (2019): 209-218. We will add this citation to the text.
>
> ### _What part of the experiment was administered by video conference? Many…_
>
> The entire experiment was administered by video conference with experimenter presence (with video off during gameplay, to avoid inadvertent reactions). Experiment presence was found to be helpful in clarifying questions that the players had on the game interface and survey questions. Given the large amount of subjective information being collected here and the relative novelty of these experiments in Hanabi, we opted for this more labor-intensive approach for our first pass in order to ensure higher data quality.
>
> ### _Where does the verbal participant commentary described on lines 240-244 come from…_
> These were comments made by subjects during the course of gameplay that were noted by the experimenter. We can make a note in Section 3.1 that experimenters made notes of subject reactions, including quotes at times.
>
> ### Regarding the study design:
> * All pairwise interaction effects were part of the mixed effects model. That is what was meant by “second-order mixed-effects models” in line 182.
>
> * Regarding summing related Likert scale responses – that is a fair point and an experiment design that we had considered. Certainly we could have grouped questions under the constructs that they were related to, but we had some hesitancy in doing so because subjects might interpret the wording of different questions under the same construct in different ways. We do not believe that the number of questions was an issue, since we performed a multiple comparison correction (lines 186-187). Combining the questions in the way that was suggested is another valid approach, but we believe that keeping them separate and using the multiple comparisons correction provides potentially more stringent statistical bounds.
>
> * On excluding discussion of “near-significance”: fair enough, though in the context of lines 203-204, it is not a (near-)significant result that affects our conclusions. It was only to say that subjects likely experienced some learning over the course of playing three games with an agent. We can remove that line.
>
> * We can report full regression tables in an updated supplement for transparency.
>
> * The paragraph on lines 222-225 is a broader comment that drives the extra analysis that was described in lines 226-233. The latter was a post-hoc comparison with specific definitions for novice and expert, so we described the expert/novice split explicitly. We can combine the two paragraphs if they seem repetitive.
>
> ### Other comments:
>
> The claims on human approaches to Hanabi came from conversations with some self-reported “expert” Hanabi players, as well as from comments from some subjects in their post-game evaluations. We are unsure of how to cite the former set of conversations, but utilizing domain knowledge expertise is certainly not unheard of for AI evaluations in games. For example, Go and StarCraft grandmasters were heavily consulted as part of the development of AlphaGo and AlphaStar, respectively. We are unaware of “professional” level Hanabi players, and unfortunately, cannot comment much further either way without breaking NeurIPS anonymization rules. Perhaps the source of these comments can be added in as a footnote in the post-anonymized phase.
>
> Lines 307-308 on participants’ reasoning simply came from grouping the Likert questions with significant inter-agent differences under the associated human factors constructs. We can make this clearer with the aforementioned change to Table 2 that ties the questions more directly back to constructs.
>
> Regarding conclusions and significance, we find it a fair criticism that the paper should address the limits of the experiment more explicitly. We will update the beginning of Section 5.3 accordingly, to focus this as a result for a single pair of rule-based and learning-based Hanabi agents. Overall, the goal was not to make the claim that these results necessarily apply broadly across all rule/learning agent comparisons, but rather that we have found a fairly stark example of this in Hanabi, and that it is an element that is worth looking out for in a domain that has thus far been almost entirely focused on score-based performance metrics. Like the reviewer has commented, we mainly hope that this work will encourage other AI researchers to give consideration to human collaboration with AI teammates beyond purely “performance” in future R&D.
>
> We suspect that similar results may be found in a game like Overcooked. For domains where rule- and learning-based agents do not perform similarly, we would expect that human reactions would be colored by both the agent behavior and the overall objective team performance. Our experiment largely rules out the latter as a significant factor for our chosen agents, which simplified our analysis. A different comparison would have to build that into their analysis.
>
> Thank you once again for the time and effort for this review.

---

> > ### Comment · Reviewer_aZxX · 2021-09-01
> > **Thank you for your response**
> >
> > Thank you for your responses to my comments, and for the opportunity to review your research. I suspect the definitions and text clarifications you’ve suggested will go a long way in enhancing the paper’s arguments -- they have already improved my understanding of the research. As before, I believe that (1) the focus on human-AI collaboration in a strategically complex game and (2) the approach of measuring performance objectively and subjectively make the overall topic suitable for NeurIPS. However, I also continue to harbor concerns about generalizability, particularly considering potential biases in the recruitment and data collection strategy.
> >
> > 1. Though unstated in the original manuscript, the study employs convenience sampling (or potentially snowball sampling) to recruit N = 29 participants. I would strongly recommend including this information in the manuscript, as sampling approach is a core detail of study design. In this case, the use of convenience / snowball sampling substantially limits the generalizability of behavioral results (see Parker, Scott, & Geddes, 2019). Various parts of the submission frame the study’s findings as generalizing to a broader population: for example, lines 11-12 of the abstract state, “We find that humans have a clear preference [...]”; lines 37-38 aver “Our results indicate that human subjects show a clear preference [...]”; and lines 310-311 of the conclusion assert “These results show that even state-of-the-art RL agents largely fail to convince humans that they are good teammates”. These statements generalize from the results at hand, discussing “humans” or “human subjects” rather than “our participants”, “Hanabi experts”, etc. Such a jump would be better justified with a participant pool like Prolific, Qualtrics, etc., which has specifically been evaluated to test representativeness. Convenience and snowball sampling can be important behavioral research tools (e.g., for qualitative research), but generally lend themselves better to pilot studies or limited exploratory claims than to general inferences or claims. For this manuscript, the type of sampling used compounds concerns I have about the overstretched / overgeneralized claims in the discussion and conclusion.
> >
> > 2. Conducting psychology or behavioral experiments over a video chat application is quite unconventional -- with the notable exception of interview studies -- due to experimenter demand effects. A more typical approach would be to hard-code instructional information through in-study tutorials and just share a link to a browser-based study. For example, from a quick search, Eger et al. (2020) were able to conduct human-agent collaboration studies on Hanabi through a browser-based design. In this case, the video-chat approach raises some substantial experimenter demand concerns (see also de Quidt, Haushofer, & Roth, 2018; Zizzo, 2010). First, were the sessions conducted by a research team member who was blind to the project hypotheses? Blinded designs are an important protection against accidentally or unintentionally "leaking" information about study predictions to participants (observer-expectancy effects; e.g., Rosenthal, 1966). Experimenters may unconsciously nudge participants toward certain choices or behaviors (e.g., preferring one agent over another). Using standardized instructions -- either through a strict script, or preferably through browser-conveyed instructions which are identical between participants -- normally help to mitigate this concern. Second, how much did participants change their behavior because (regardless of whether there were any "leaky channels" about the research predictions) they knew they were being watched? Solely the knowledge that they are being watched by someone can change a participant's behavior (the Hawthorne effect, panopticon effects, etc.). It's notable to me that at least two participants spoke in complete, explanatory sentences (lines 242-243), which fits more with communication-oriented speech than frustrated (emotive) utterances.
> >
> > 3. On scale combination: I agree that your approach was sufficiently conversative, in terms of controlling the family-wise error rate -- though this is true because of the specific correction method applied (Holm-Bonferroni), rather than any multiple comparisons correction. Many correction methods assume statistical independence between the tests, which would be a faulty assumption in this case. In any case, on the conceptual side, I would encourage you to look into item response theory, the subfield within psychometrics that focuses on theoretical and technical aspects of measuring attitudes, beliefs, etc. In particular, I think it’s worth considering what latent variables are intended to be measured, whether those latent variables are accurately captured by single responses to individual questions, and how they might be better estimated through combining multiple observations stemming from a set of questions (an “instrument” in psychology).
> >
> > 4. Regarding the inclusion of expert knowledge of Hanabi in the manuscript: research papers often discuss information that is not common knowledge, and typically are expected to attribute the source of that information. This source attribution is normally either handled by a citation, or is provided directly from the paper’s data and analysis itself. If the authors of a paper are experts in a game being studied, I think it’s fine (and important) to include the statement “From our knowledge of the game [...]”, if that is the source of the information at hand. This avoids implying that the information is common knowledge, and allows readers to either accept or scrutinize the information as they feel appropriate. Alternatively, in this particular case, if the goal is to introduce new data from your human-AI teaming experiments -- that requires transparency in describing how the data were gathered, again so that readers can scrutinize potential sources of bias. For example, the methods could clarify that there were post-study debriefing interviews, and then the discussion can explicitly state “In the debriefing interviews, participants frequently stated that pre-game convention-setting and post-game reviews are common practices for human Hanabi players [...]”.
> >
> > I think the present work identifies important hypotheses and contributes value for the ML research community. NeurIPS submissions are held to very stringent standards for the machine learning methods and techniques they use; I think it's just as important that we apply similar standards to the behavioral research methods and techniques brought to our studies.

---

> > > ### Author Response · Authors · 2021-09-08
> > > **response to reviewer aZxX**
> > >
> > > Thank you for the comments.
> > >
> > > 1.	The clarification on sampling method is valid, and is something we will include more explicitly in the text. Additionally, we will change the language in our discussion to be more conservative in order to respect the limitations of generalizability given the experiment. We would note though, that snowball and targeted-demographic sampling is commonly utilized by other Hanabi human-AI experiments, including Eger et al, 2020 (_“For both experiments, we recruited participants via snowball sampling from various social media, such as twitter, facebook, boardgamegeek and the board game and game AI subreddits”_) and Liang et al. 2019 (_“We recruited participants for our user study through online forums, namely from Reddit (r/boardgames and r/cardgames),and linked them to our online Hanabi implementation.”_). Such recruitment is more necessary in the Hanabi setting than in general gameplay or psychology experiments due to the specialized skillset required (Hanabi expertise), lest our subject pool be overwhelmingly comprised of novices. So we agree that the sampling concern is valid, though it is a constraint we have in this type of experiment.
> > >
> > > 2.	The experiment used a standardized set of initial instructions, presented through a set of slides. The experimenters were aware of the general hypotheses being tested. To avoid some of the experimenter influence, experimenters turned off their video during much of the experiment (and always during the games), and were generally muted except when introducing the experiment, introducing the surveys, providing clarification, and at the end of the experiment. Subjects did know that they were being watched, which likely explains the occasional use of full sentences (that were recorded by the experimenters), but there were also free-response sections of the surveys in which subjects tended to use full sentences as well. We believed that it was important for experimenters to be involved “live” in some fashion due to our inability to do audio/visual recordings (due to far more stringent IRB restrictions and the remote nature of the experiment), and COVID-related restrictions meant that the experiment did have to be fully remote and we could not simply be in the next room, per se. Additionally, the presence of a human experimenter for this virtual experiment removed a fair amount of the ambiguity behind how much of the data came from unique subjects, which was an issue in the aforementioned other Hanabi human experiments.
> > >
> > > 3.	Thank you for the suggestion to look into IRT. There were certainly groupings of questions that stem from the same human factors constructs (the latent variables of interest), which is information we intended to add to Table 2. Perhaps that would clarify the relationship between the Likert scale statements. However, it would seem that taking the approach of measuring responses to specific questions individually, rather than in groups, while maintaining the multiple comparisons correction, would be more conservative than using the instrument approach? IRT would be more statistically powerful, and perhaps what we have done is overly conservative, but that might lend additional credence to the significance of the current results (within the confines of the sampling method).
> > >
> > > 4.	Thank you for the suggestions. We can clarify in the text when the source of information was our own game knowledge and when it was described by subjects post-game.
> > >
> > > We hope we have clarified some of our procedures in the context of the related literature and allayed some of your concerns. Thank you once again for the insightful comments.

---

### Official Review · Reviewer_Tigt · 2021-07-21

**Rating:** 5
**Confidence:** 4

**Summary:**

The work investigates tradeoffs between objective metrics of performance and subjective metrics of partner collaborative-ness in human-AI cooperation scenarios, taking the Hanabi domain as a case study. After reviewing previous work in this domain, the authors then perform a detailed user study with two state-of-the-art human-AI bots (one rule-based, one learned), and demonstrate that interestingly, users tend to prefer rule-based rather than learned Hanabi agents (even though they have similar objective performance metrics). They argue that this has important implications for the field of AI, as we want AI agents to be considered trustworthy teammates in addition to obtaining high reward.

**Limitations And Societal Impact:**

If the intention of the authors was to claim that rule-based agents in Hanabi tend to be perceived as better agents relative to learned ones (as it seems in various parts of the paper), doing a comparison with just one rule-based and one learned agent seems a bit limiting. While it seems plausible that learned approaches tend to systematically be less interpretable to human partners, doing a comparison with just one member of each group (rule vs. learned) seems potentially insufficient: there might be other factors that are method-specific which influence the perception of the user.

In particular, I would not expect all rule-based systems to be better than all learned systems. If the rules are not human-like or not very good, trivially such a system would not be well-perceived by users. On the other hand, the untrustworthiness (and convergence to obscure equilibria) of learned AI agents doesn't seem to necessarily be an intrinsic characteristic of learned systems: if one trained AI agents which leverage human data for learning policies which complement users' policies, that would likely lead to much better perception of such agents (this phenomenon has been shown at least when comparing to pure self-play trained collaboration partners, e.g. in [9] from your paper).

Further qualifying the claim, reducing the generality of it, or addressing why comparing just one rule-based and learned agent is enough would strengthen the paper.

**Main Review:**

## Originality

While there is no new method presented in the paper, the main contribution would consist of the in-depth analysis of a user study performed to assess 1) the performance of two difference categories of human-AI Hanabi agents and 2) how they are perceived by the users. While minor user studies were performed for individual agents (focusing on reward), I am not aware of any studies comparing different Hanabi AI bots or focusing on the qualitative aspects of the behavior. Generally, this result is interesting and provides further an in depth account of the phenomenon that by default, learned agents for collaboration will not be human-interpretable even if they perform very well with each other (and perform ok with humans). In particular, I don't know of any work showing that this holds even when comparing to rule based agents, which might tend to have more baked-in inductive biases which lead them to be more human-interpretable.

## Quality / Clarity

The claim that people tend to prefer the SmartBot Hanabi AI agent relative to the Other-Play one is well supported by the statistical analysis, which seems thorough and well executed. The difference between the perceptions of novice and experienced players was particularly interesting – this seems like something worth spending more time on explaining. It would also be interesting to think about whether one should expect similar types of trends in other domains than Hanabi.

Generally, the paper is very clearly written, well organized, and a pleasure to read. The bar charts in particular were well made and very informative.

## Significance

While the analysis was clear, I found the takeaway of this work to be less so: are the authors simply showing that some agents are perceived to be less cooperative than others, even under similar levels of objective performance (311-312)? If so, it would be interesting to head what preliminary ideas for including these subjective judgements into the AI generation process would be. Alternatively, if the intention of the authors was to claim that rule-based agents in Hanabi tend to be perceived as better agents relative to learned ones (as seems to be suggested in various parts of the paper), doing a comparison with just one rule-based and one learned agent seems a bit limiting (more of this in the limitations below). Another potential takeaway might be that the authors are suggesting that one should use rule-based agents because they tend considered better partners (although this is not explicitly said anywhere in the paper)?

Generally, it would be helpful to spell out more obviously what the intended contribution is.

**Time Spent Reviewing:**

3

---

> ### Author Response · Authors · 2021-08-10
> **initial response to reviewer Tigt**
>
> Thank you for the review and the suggested changes.
>
> Regarding the quality/clarity comments – we can add additional commentary in Section 4.2, 4.3, or 5.2 to discuss the subject experience differences. We have discussed potential generalizations beyond Hanabi in Section 5.3, though we do want to be careful not to overextend our conclusions, giving the limitations of the experiment.
>
> With regards to the significance of the paper, our intended contribution was simply to test the hypothesis: “In the collaborative game of Hanabi, state-of-the-art learning-based agents will outperform the best hand-coded rule-based agents when teamed with human players _and therefore_ be preferred by human teammates.” This hypothesis was motivated by the recent success of learning-based agents in competitive environments (e.g. Go, StarCraft) and we wanted to see if such success extended to human-collaborative environments. Our experiments unexpectedly refuted this hypothesis which we found to be an interesting result worth publication. Since we did not propose our own novel agent, we had no “horse in the race” and we were able to take a more objective perspective during experiments without risk of biasing experiments to favor some agent of our own. In short we did not “intend” to show the superiority of learning or rule-based agents, we simply had a hypothesis that learning-based would be better; this turned out not to be the case. For clarity, we can make our hypothesis more explicit in the introduction (lines 32-40).
>
> We believe that our contributions are more general than simply showing that some agents are perceived to be less cooperative than others. Specifically, lines 308-312 point out the lack of consideration for human perception of teammates in the literature of the current push to use Hanabi as a multi-agent RL platform. Our results point to a specific consequence in this paper, using state-of-the-art agents from the rule-based and RL classes of AI as an illustration where humans strongly disliked the RL agent in comparison to the rule-based agent, with the key takeaway being that human perceptions of teammates must be considered in settings where human-AI teaming is an eventual goal. We can make this point clearer here and at the end of the introduction as well.
>
> As far as utilizing only two specific agents for our tests, we note that many of the AI bots from the literature do not have publicly-available implementations, as mentioned in the limitations of our work (Section 5.3). We did perform some preliminary testing with other learning-based agents (from [24]), but they 1) performed much worse than OP, and 2) were much more strongly disliked by human teammates than OP. From this, we concluded that it would not add much to the results to use a wider variety of learning agents, when OP appears to already be an example of best-in-class for cross-play.
>
> We can provide additional commentary for our agent selection in Section 3.1.
>
> The reviewer makes a fair point about clarifying and more clearly delineating the limits of what may be concluded from this work. We will describe these limitations more explicitly at the beginning of Section 5.3.
>
> Thank you again for the time and effort for this review.

---

### Decision · Program_Chairs · 2021-09-28

**Decision:**

Accept (Poster)

**Comment:**

Reviewers found the problem setting and focus on subjective agent preferences interesting and promising, but ultimately I don't believe the paper is ready for publication at NeurIPS. Reviewers raised concerns about 1) the generalizability of the findings given the focus on one learning agent and one bot in one environment, 2) the human evaluation protocol (especially recruitment and the single-blind setup), and 3) a lack of qualitative analysis backing up the quantitative findings. Although the author response to #2 was reassuring and #3 could potentially be addressed before publication, #1 ultimately limits the interest of the current work to the community. As suggested by the reviewers, the authors could increase the scope of their findings by e.g. including a behaviorally cloned agent, including another collaborative environment such as Overcooked, or including enough qualitative analysis that it becomes more clear where the results should and should not generalize.


**Consistency Experiment:**

NeurIPS has a long history of experimentation. In 2014, NeurIPS ran an experiment in which 10% of submissions were reviewed by two independent committees to quantify the randomness in the review process. This year, we repeated a variant of this experiment to see how the quality of the review process has changed over time.  This paper was part of the experiment and was therefore assigned to two committees (consisting of reviewers, an Area Chair, and a Senior Area Chair) that reached independent decisions.  If both committees made the same recommendation, this recommendation was followed. If a single committee recommended acceptance, the paper was accepted (with the exception of a few cases in which the other committee identified what we considered a fatal flaw, e.g., an error in a key result).

This copy’s committee reached the following decision: **Reject**

The other committee assigned to the paper recommended **Accept (Poster)**.  You can find the other set of reviews, along with any follow up discussion with the authors here:
https://openreview.net/forum?id=x_JOyw5CLP